# A massively parallel screening platform for converting aptamers into molecular switches

Alex M. Yoshikawa [1,6], Alexandra E. Rangel[2,6], Liwei Zheng [2], Leighton Wan [3], Linus A. Hein[4], Amani A. Hariri[2], Michael Eisenstein[2,4] & H. Tom Soh [2,4,5] ✉

Aptamer-based molecular switches that undergo a binding-induced conformational change have proven valuable for a wide range of applications, such as imaging metabolites in cells, targeted drug delivery, and real-time detection of biomolecules. Since conventional aptamer selection methods do not typically produce aptamers with inherent structure-switching functionality, the aptamers must be converted to molecular switches in a post-selection process. Efforts to engineer such aptamer switches often use rational design approaches based on in silico secondary structure predictions. Unfortunately, existing software cannot accurately model three-dimensional oligonucleotide structures or non-canonical base-pairing, limiting the ability to identify appropriate sequence elements for targeted modification. Here, we describe a massively parallel screening-based strategy that enables the conversion of virtually any aptamer into a molecular switch without requiring any prior knowledge of aptamer structure. Using this approach, we generate multiple switches from a previously published ATP aptamer as well as a newly-selected boronic acid base-modified aptamer for glucose, which respectively undergo signal-on and signal-off switching upon binding their molecular targets with second-scale kinetics. Notably, our glucose-responsive switch achieves ~30-fold greater sensitivity than a previously-reported natural DNA-based switch. We believe our approach could offer a generalizable strategy for producing target-specific switches from a wide range of aptamers.

Nature has evolved a diverse toolbox of nucleic acid-based molecular switches—known as 'riboswitches'—that enable living organisms to sense and respond to environmental stimuli. Riboswitches are complex folded RNA domains that control gene expression via allosteric structural changes triggered by binding to a specific ligand[1,2]. For example, the bacterial glycine riboswitch facilitates glycine breakdown by controlling the expression of three genes required for degradation in response to glycine binding[3]. Similar target-responsive RNA- and

DNA-based molecular switches have potential utility in a variety of technological applications, and researchers have engineered a number of synthetic nucleic acid-based constructs that mimic naturally-occurring riboswitches and undergo similar binding-induced conformational switching. In some cases, these are used to trigger the same kinds of gene-regulatory functions as occur in nature[3], but engineered riboswitches are also utilized as molecular switches in biosensing applications[4]. Such biosensors are typically based on

[1]Department of Chemical Engineering, Stanford University, Stanford, CA 94305, USA. [2]Department of Radiology, Stanford University, Stanford, CA 94305, USA. [3]Department of Bioengineering, Stanford University, Stanford, CA 94305, USA. [4]Department of Electrical Engineering, Stanford University, Stanford, CA 94305, USA. [5]Chan Zuckerberg Biohub, San Francisco, CA 94158, USA. [6]These authors contributed equally: Alex M. Yoshikawa, Alexandra E. Rangel. ✉e-mail: tsoh@stanford.edu

aptamers that undergo a reversible structure-switching mechanism, which is then coupled to either an optical or electrochemical readout. These engineered nucleic acid molecular switches have been used for a wide range of applications such as imaging metabolite dynamics in living cells[5,6], real-time monitoring of drug molecules in live animals[7,8], and targeted cancer therapy[9–11].

However, it remains a challenge to generate aptamer switches, because most aptamers assume a stably folded structure and do not undergo a binding-induced conformation change. The majority of aptamer switches are therefore created via a post-selection engineering approach—for example, converting the aptamer into a molecular beacon, split-aptamer construct, or intramolecular strand-displacement reagent[12]. However, these approaches all rely on rational design, and therefore require prior knowledge of the aptamer structure and entail careful balancing of thermodynamic states. Such detailed structural characterization has only been achieved for a relatively small number of aptamers, and most design efforts rely on in silico predictions of secondary structure. However, even the most advanced modeling software often fails to account for non-canonical base-pairing motifs, such as G-quadruplexes and pseudoknots, and such structural elements are often critical to the function of both natural riboswitches and synthetic aptamer switches[13]. These predicted structures also typically cannot capture the three-dimensional folding of aptamer switches, which can be highly relevant to target recognition and binding. As a consequence, the initial aptamer switch design effort is often followed by a time-consuming trial-and-error process in which multiple constructs are fabricated and evaluated. To overcome these obstacles, specialized methods have been developed for the direct screening of structure-switching aptamers, such as capture-SELEX[14,15]. Here, a solid support is modified with a short complementary DNA helper strand that hybridizes to the aptamer library in the absence of target, and which enables partitioning of sequences that undergo target binding-induced dissociation from the solid support as a result of undergoing a conformational change. While this approach has proven successful, considerable effort is required to perform the selection, as evidenced by the relatively small number of aptamer switches in the literature[16–23].

In this work, we describe a massively parallel screening strategy that could greatly accelerate the development of target-responsive molecular switches from existing aptamers without any a priori structural knowledge, eliminating the need for computer modeling or design heuristics. To achieve this, we build upon our recently developed non-natural aptamer array (N2A2) system, in which large numbers of natural DNA or base-modified aptamers are synthesized directly on the flow-cell of a modified Illumina MiSeq instrument and characterized in a massively parallel manner[24,25]. This allows us to rapidly screen as many as ~1 million different switch scaffolds in a single experiment, rather than iterating through multiple cycles of design, synthesis, and evaluation. The screening library consists of an array of anchored displacement strand (ADS) switch constructs, in which the aptamer of interest is coupled to a library of different switching strands with a variable 'switch domain' sequence. The screening process identifies those switching strand sequences that can efficiently hybridize to the aptamer in the presence of the target, but which become separated when target binding causes the aptamer sequence to assume a fully folded conformation.

We initially test our approach with a natural DNA-based ATP aptamer and demonstrate the capacity to directly observe and identify thousands of fluorescent aptamer-based molecular switches. Interestingly, secondary structure prediction software fails to predict correct folding for many of the switches we discover, highlighting the importance of non-canonical interactions to the structure and function of aptamer switches. We then use the same strategy to identify "non-natural" aptamer switches containing chemically-modified bases. We first select a boronic acid-modified aptamer that binds glucose, and

then use our screening platform to create multiple high-affinity molecular switches that respond to glucose, surpassing the sensitivity of previously reported glucose aptamers[26]. Importantly, these switches exhibit a minimal decrease in affinity compared to the parent aptamer. These results demonstrate that our platform should offer a generalizable strategy for converting aptamers to target-specific molecular switches.

## Results and discussion

### Strategy for high-throughput selection of aptamer switching domains

Our screen draws inspiration from competition-based aptamer switch designs[27], however, in our strategy the aptamer and displacement strand are coupled together via base-pairing to form an ADS construct (Fig. 1a). The 'aptamer strand' consists of a known aptamer of interest, which is labeled at its 5' end with a fluorescence quencher group and flanked at its 3' end with an anchor sequence. The 'switching strand' is the variable component of the screening process, and is responsible for endowing the construct with the ability to undergo a target binding-induced conformational switching that results in a change in fluorescent signal. The switching strand comprises a sequence complementary to the aptamer strand's anchor sequence, a poly T linker, and a randomized 10-nucleotide switch domain (SD) sequence, and is fluorescently labeled at its 3' end.

In a switching-compatible ADS construct, the complementary anchor regions of the two strands are hybridized, and the SD sequence also interacts with the aptamer sequence in the absence of target. The SD sequence is attached to the anchor strand through a 25-nucleotide poly-T linker, which we believed would have minimal interaction with other DNA strands in the absence of extended stretches of adenosine bases. The length of the linker is crucial; too short of a linker may prevent the SD from accessing the entire aptamer due to spatial constraints, and too long of a linker may result in slower kinetics and the requirement of longer SD sequences due to a decreased effective concentration of the SD relative to the aptamer[27]. Although in many cases the ADS constructs will not change signal in response to the target molecule, in successful ADS constructs, target binding causes the aptamer strand to undergo a conformational switch that changes the average distance between the fluorophore and quencher, resulting in altered fluorescent signal (Fig. 1b).

In some cases, this conformational change will produce a 'signal-on' readout with increased fluorescence from target binding, whereas other constructs will undergo a 'signal-off' switching where fluorescence is more strongly quenched. Our approach bypasses the time-consuming and expensive process of structural analysis, rational design, and optimization by simply screening nearly every possible N10 SD sequence (~1 million) in a single assay. This makes it possible to evaluate the entire sequence space and identify optimal switch constructs without requiring prior knowledge of the structure or binding sites of the aptamer. However, longer SD regions (>10 nucleotides) that interact more strongly with the aptamer could also be used in this screening process.

The entire ADS screening process takes place on a MiSeq flow-cell, and involves three steps: 1) sequencing of the switching strand library, 2) assembly of the ADS aptamer constructs, and 3) identification of target-responsive aptamer switches (Fig. 1c). This allows us to directly link the genotype of each switching strand sequence to a functional phenotype (i.e., switching behavior) for the resulting ADS construct. In the first step, the switching strand library with a variable N10 SD region is sequenced on the MiSeq using standard Illumina sequencing protocols. We then assemble the ADS constructs on the surface of the flow-cell. Briefly, this entails targeted cleavage of the sequencing primer-complementary sequence adjacent to the randomized SD domain via a DdeI restriction enzyme site incorporated into the library molecules, after which the library clusters are fluorescently labeled

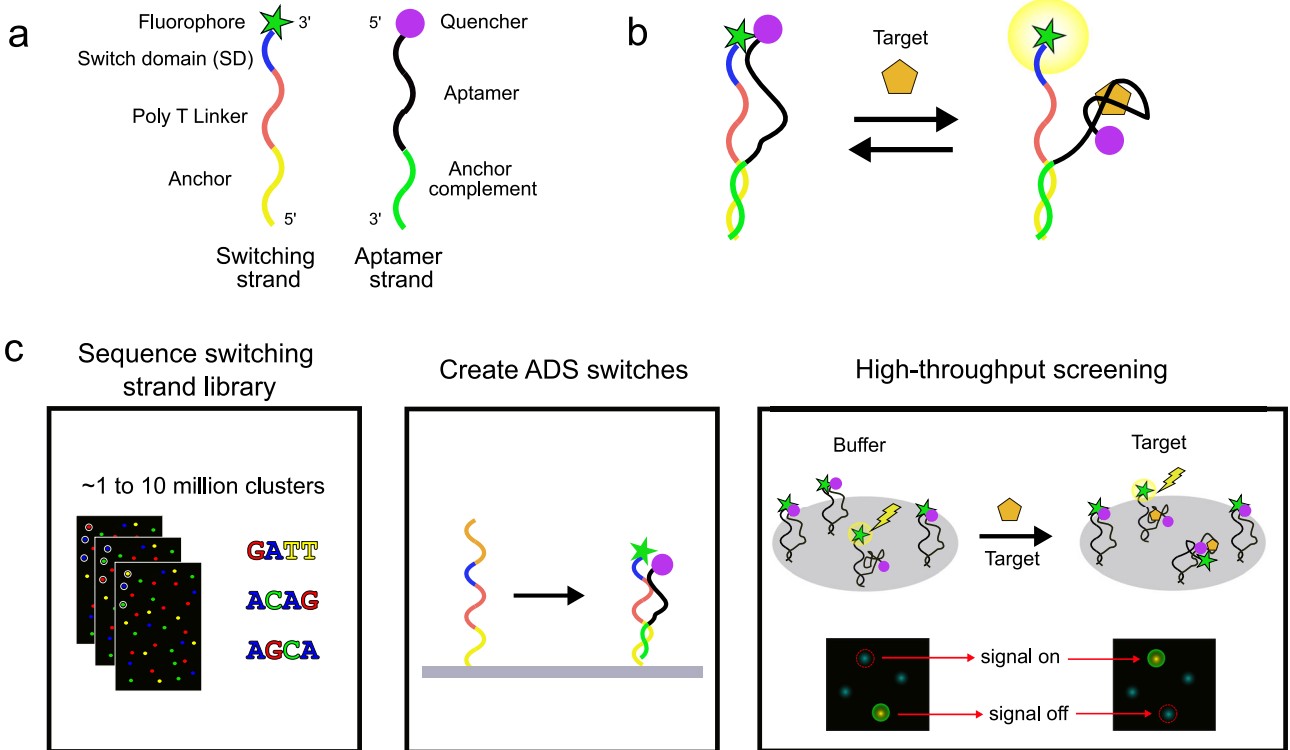

**Fig. 1 | Overview of the ADS construct and the high-throughput screening process used to convert known aptamers to molecular switches. a** Design of the fluorophore-labeled switching strand and quencher-labeled aptamer strand. **b** Target-induced conformational changes in the ADS construct result in a change in distance between the fluorophore and quencher, providing an optical readout. **c** Overview of the screening process. First, the switching strand library is sequenced

on the flow-cell. Second, the ADS constructs are assembled on the surface of the flow-cell via addition of aptamer strands. Lastly, target-responsive molecular switches are identified by sequentially imaging the flow-cell in buffer alone and with the target molecule. Imaging data from each ADS construct cluster reveals the presence of switches for which target binding results in increased (signal-on) or decreased (signal-off) fluorescence.

with Cy3 using a terminal deoxynucleotidyl transferase (TdT) enzyme (see Methods and Supplementary Fig. 1 for a detailed description of this step). The ADS constructs are then fully assembled by annealing the quencher-tagged aptamer strands onto the switching strand clusters. In the third and final step of the screening process, the flow-cell is washed with buffer and a fluorescent image of the flow-cell is acquired. The target molecule is then injected onto the flow-cell, and another fluorescent image is taken. By comparing the intensity of these two images, we can identify ADS constructs that undergo a target-induced change in fluorescent signal, whether signal-on or signal-off. The flow-cell is then washed with 50 mM NaOH to remove the target and aptamer from the ADS library clusters, after which the aptamer strand is re-annealed and replicate measurements are performed to identify outliers and characterize the variance in the experiment.

## Screening ATP-responsive ADS constructs

As an initial test of our high-throughput ADS screening platform, we used a well-studied DNA aptamer that binds ATP with a dissociation constant ($K_D$) of 6 µM[1,28] (Supplementary Table 1). This aptamer has been extensively studied, and unlike most DNA aptamers, its three-dimensional structure has been solved via NMR studies[29], thus enabling us to better evaluate the nature of the interactions between the screened switching strands and the parent aptamer itself. Before proceeding with screening, we first validated the performance of the enzymatic steps involved with ADS synthesis on magnetic beads via flow cytometry (see Methods for details). These results confirmed successful enzymatic cleavage of primer-binding regions (Supplementary Fig. 2) and successful enzymatic coupling of the fluorescent label (Supplementary Fig. 3). We then proceeded to set up the high-throughput ADS screen on our N2A2 platform using the ATP aptamer. After sequencing the switching strands, the ADS constructs were

assembled on the surface of the flow-cell. Examination of the flow-cell images confirmed that the switching strands were selectively labeled with Cy3 (Supplementary Fig. 4). The quencher-labeled aptamer strands were then hybridized onto these clusters, completing the formation of the ADS construct. The clusters were imaged in alternating cycles of buffer and 500 µM ATP, with five of each cycle performed in total. The cluster intensities calculated by the MiSeq software were linked to each cluster using a custom Python script. To eliminate false positives and reduce noise, clusters with signal below 100 RFU and above 1000 RFU were filtered and discarded. We have found that clusters with low initial fluorescence are noisy and that clusters with intensities >1000 RFU are typically caused by inaccurate image processing or by unwanted particles, such as dust, on the flow-cell. To eliminate data with high levels of variance, we also excluded clusters that had over 30% relative standard deviation (RSD) between either replicate buffer cycles or replicate ATP cycles. Finally, the switching domain clusters were sorted by the ratio of the signal in ATP compared to buffer.

Analysis of the filtered data revealed thousands of ATP-responsive aptamer switches. While the screen can identify both signal-on and signal-off switches, the ATP aptamer yielded primarily signal-on aptamer switches. Of the ~455,000 clusters, over 8000 exhibited at least a two-fold change in fluorescence intensity after the addition of ATP. The top 1000 unique SD sequences showed a more than four-fold increase in fluorescent intensity (Supplementary Fig. 5), with an eight-fold increase in intensity from the top-performing switch. To investigate whether particular regions of the ATP aptamer were preferentially interacting with SD regions in these switches, we created a histogram of the frequency with which each base-position in the ATP aptamer ended up being recognized by an SD sequence in the top 1000 ADS switch candidates (Fig. 2a). We observed two clear peaks within the

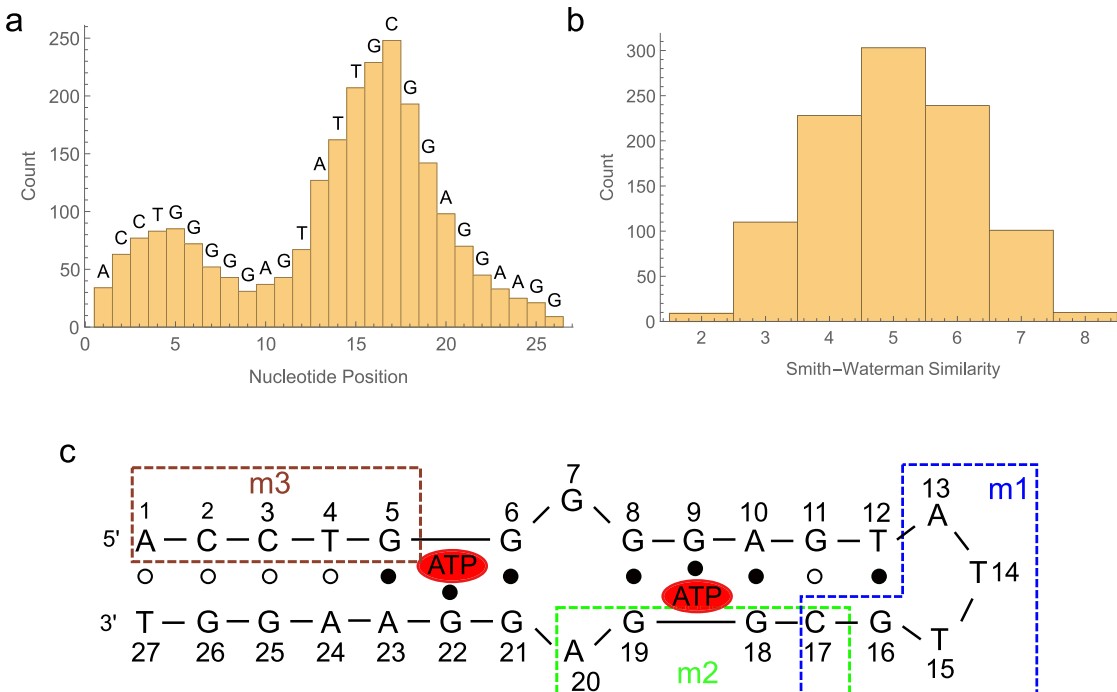

**Fig. 2 | Analysis of the 1000 best-performing ADS sequences. a** Histogram of the frequency with which each base-position within the ATP aptamer was complementary to an SD sequence. This analysis was based only on the longest complementary region for each SD, with complementary regions of <3 nucleotides discarded. Nucleotides at each position in the aptamer are labeled above the histogram. Source data are provided as a Source Data file. **b** Histogram of Smith-Waterman similarity distances between the ATP aptamer and the reverse-complement of the top 1000 SDs from our screen. Source data are provided as a Source Data file. **c** Secondary structure of the ATP aptamer as previously discovered via NMR[29]. Boxed regions m1, m2, and m3 indicate segments complementary to the three recurring SD motifs that we identified.

ATP aptamer sequence that were preferentially targeted for base-pairing by SD sequences. To our surprise, this sequence alignment analysis revealed that many of these SD regions had minimal complementarity with the aptamer (Fig. 2b). Indeed, more than 10% of the top 1000 SD regions had a Smith-Waterman similarity score of 3 or less to the reverse-complement of the aptamer, meaning that only 3 of the 10 bases in the SD would be predicted to hybridize to the ATP aptamer. This leads us to believe that that many of these constructs must rely on extensive mismatch base-pairing or non-canonical nucleic acid interactions, which is in keeping with previous findings that naturally evolved riboswitches also extensively utilize such non-canonical interactions[1].

We determined that the two regions of the ATP aptamer that were preferentially targeted by the switching domains corresponded to distinct recurring sequence motifs by analyzing the top sequences using the motif discovery tool in the MEME suite (version 5.3.3)[30] (Supplementary Fig. 6). We found three sequence motifs (m1, m2, and m3) that exhibited a statistically significant association with switching function (E-value <0.05). Motifs m1, m2, and m3 were respectively represented in 28.2%, 9.3%, and 4.9% of the top 1000 sequences. All three were complementary to different regions of the ATP aptamer (Fig. 2c) and overlapped with one of the two histogram peaks identified in Fig. 2a. By analyzing the secondary structure of the ATP aptamer as determined by NMR[29], we found that the m2 motif was complementary to one of the ATP binding pockets (nucleotides 17-20), whereas the m3 motif recognizes the 5′ end of the aptamer stem (nucleotides 1-5)—both representing regions that one might target with a rationally-designed SD. Interestingly, the m2 motif contained a mismatch in the SD sequence, indicating that imperfectly complementary displacement strands may yield optimal performance in some switch constructs. This is not surprising, given that mismatches can finely tune the thermodynamics of hybridization reactions. However, rationally designed displacement strands typically do not contain

mismatches, and it is therefore likely that this subset of sequences would have been overlooked with such an approach. We were also surprised to see that the m1 motif (nucleotides 13-17) was so abundantly represented—being present in more than a quarter of the top sequences—and this indicates that the short DNA loop recognized by m1 may be a particularly responsive target for the development of displacement strand-based switches. This is contrary to conventional wisdom, which suggests that such strands should be designed to target the binding pocket or the stem that stabilizes the aptamer structure in order to ensure that the aptamer cannot bind the target without first decoupling from the displacement strand. These results highlight the value that can be derived by employing a broader and more agnostic approach to identify displacement strands that confer optimal switching properties, rather than relying on potentially misleading a priori assumptions about structure-function relationships.

**Experimental validation of ATP aptamer switches**
To verify the function of the aptamer switches, we chose a subset of eight individual ADS constructs for characterization and further analysis (Supplementary Table 2). All eight sequences displayed a large—ranging from 6- to 8-fold—increase in fluorescence intensity when bound to ATP during the ADS screen, indicating a robust signal-on response (Fig. 3a). We chose the four that displayed the greatest percent change between the buffer and target cycles (atp-1–4), as well as the top-performing sequence with each of the three SD motifs identified above. Atp-4 was also the top sequence containing the m-3 motif, whereas atp-5 and −6 were respectively the best-performing sequences containing the m-1 and m-2 motifs. Finally, we selected two sequences (atp-7 and −8) that displayed low complementarity (three nucleotides) with the aptamer, but nevertheless exhibited a strong fluorescence increase in response to ATP binding. Since the screen relies on cluster intensity values that are automatically extracted from the raw images by the MiSeq software, we further validated that these were

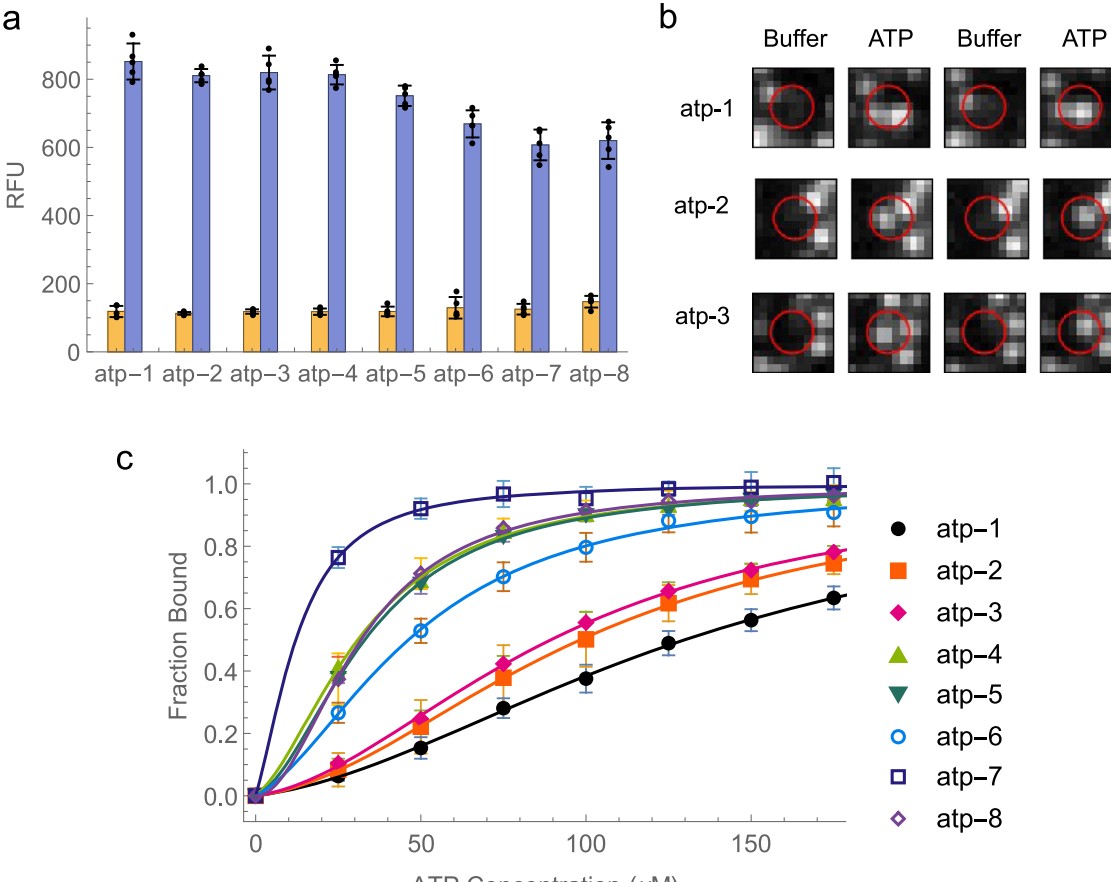

**Fig. 3 | Identification and characterization of ATP aptamer switches. a** Results from the high-throughput screen of switching domains for the ATP aptamer. Orange and blue bars respectively represent the cluster intensity in buffer and 500 μM ATP. Data are presented as mean values (*n* = 5 replicates) and error bars represent the standard deviation of the measurements. Individual data points are overlayed as black dots. Source data are provided as a Source Data file. **b** Extracted images of individual ATP-responsive ADS clusters (red circle) on the MiSeq flow-cell from multiple buffer and ATP cycles. **c** Validation of selected ADS constructs

identified from the high-throughput screen. 50 nM fluorophore - and quencher-labeled ADS construct was incubated with various concentrations of ATP and measured on a plate reader (*n* = 4 replicates). A two independent binding site model was used to fit the raw data and normalize the binding signals between 0 and 1, as represented by the solid lines. Data are presented as mean values and error bars represent the standard deviation of the measurements. Source data are provided as a Source Data file.

true-positive responses by manually inspecting the original cluster images from the flow-cell using a custom Python script (representative cluster images shown in Fig. 3b). While the N2A2 system is a powerful tool for screening many aptamer sequences in parallel, the instrument is not necessarily designed for quantitative fluorescent intensity measurements. Rather, the MiSeq hardware is intended to differentiate between four fluorophores (corresponding to each DNA base) during conventional sequencing. Measurements collected from this instrument can potentially be confounded by factors such as avidity effects between neighboring ADS constructs within a cluster, or artifacts introduced by the algorithm that is being used to convert the cluster images into relative fluorescent units (for example, the algorithm may pick up signal from nearby clusters). This means that the screening step is best suited to provide relative binding values between the ADS clusters and to identify the best switches for further characterization.

We therefore sought to validate and characterize the activity of the aptamer switches using a plate-reader assay. We chemically synthesized the selected switching strands and assembled fluorophore- and quencher-modified ADS constructs by annealing the switching and aptamer strands at a 1:1 ratio. We then measured the target-induced fluorescence response of a fixed concentration (50 nM) of these ADS constructs on a plate reader after titrating with various concentrations of ATP (Fig. 3c, Supplementary Fig. 7).

Each construct displayed a dose-dependent fluorescence increase, in agreement with the results from the high-throughput screen, and we determined their $K_D$ by fitting to a two-binding site model that has been previously used to characterize ISD constructs developed from the same aptamer. No dose-dependent change in fluorescent signal was observed for five control ADS constructs (atp1-s1, atp1-s2, atp2-s1, atp4-s1, and atp6-s1) containing scrambled switching domains (Supplementary Fig. 8), indicating that the specific sequence of the switching domain was necessary to enable successfully binding-induced switching. $K_D$ values ranged from 12–157 μM, which is reasonable given that the original ATP aptamer has a $K_D$ of 6 μM. It is worth noting that the highest-affinity construct we tested was atp-7, which was chosen based on its low degree of complementarity to the ATP aptamer—this again highlights the fact that the determinants of an effective SD for an aptamer switch might be somewhat counter-intuitive based on current design principles. Although atp-7 had the highest affinity, it also had a smaller total signal change when compared to some of the other ADS constructs, such as atp-6. Since the signal change is dictated by the distance between the fluorophore and quencher in the target-bound and target-unbound states of the ADS construct, it is possible that these dye locations could be optimized post-screening to increase the total signal change. We also examined the binding kinetics of three ADS constructs (atp-4, atp-5, and atp-6) using time-

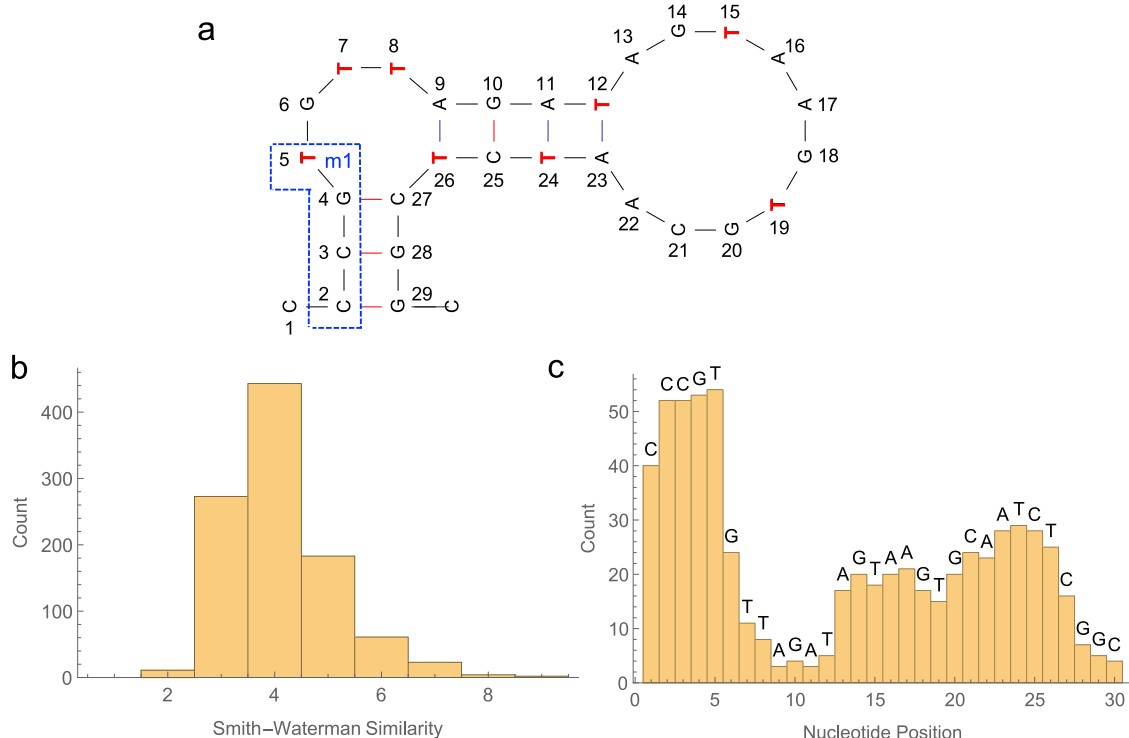

**Fig. 4 | Analysis of the top 1000 unique glucose SD sequences. a** Predicted secondary structure of the phenylboronic acid-modified glucose aptamer NNGmin. Red bolded Ts denote location of modifications. The boxed region m1 indicates a motif that was highly recurrent among the sequence elements targeted by our top SDs. **b** Histogram of the frequency with which each aptamer base-position was complementary to an SD. Source data are provided as a Source Data file. **c** Histogram of Smith-Waterman similarity distances between the glucose aptamer and the reverse-complement of the top 1000 SD sequences. Source data are provided as a Source Data file.

course fluorescence experiments on a plate-reader (Supplementary Fig. 9). Since the switching strand can act as a competing binding partner to the aptamer, it is reasonable to expect that both the binding kinetics and affinity will decrease relative to the parent aptamer. The kinetics of the ADS constructs were indeed slightly slower than the parent aptamer, reaching equilibrium in <10 seconds rather than <5 seconds[31], but this small decrease in binding kinetics is unlikely to have an impact on most applications in which aptamer switches could be utilized.

### A base-modified DNA aptamer switch for glucose

We next explored whether our method could be generalized to aptamer sequences containing non-natural, chemically-modified nucleic acids in order to identify high-affinity aptamer switches responsive to glucose. Small-molecule targets such as glucose have proven to be challenging targets for natural DNA and RNA aptamers—in part because of the limited number of chemical functional groups they offer as recognition sites[32]. Our group and others have previously demonstrated that aptamers that incorporate chemically-modified nucleobases can achieve robust binding to such challenging targets[33], and we therefore set out to isolate such an aptamer as the foundation for a glucose-responsive switch[27].

To this end, we first employed our previously published click-PD screening method[34] to generate a boronic acid-modified aptamer that achieves high affinity for glucose. Briefly, click chemistry is used to covalently couple alkyne-labeled chemical modifications onto libraries of monoclonal aptamer particles (i.e., particles containing copies of a single aptamer sequence) displaying sequences that incorporate azide-tagged non-natural uracil nucleotides. Fluorescence-activated cell sorting (FACS) is then used to interrogate the binding of these base-modified aptamer particles to a labeled target, and those that exhibit high fluorescence—and thus high affinity—can be individually sorted in a high-throughput manner. We chose phenylboronic acid as the base modification for this selection, as it forms stable cyclic boronate esters with saccharide diols, and should therefore facilitate the generation of high affinity aptamers to glucose. We also appended an ortho-aminoethyl moiety onto the phenylboronic acid modification used in our selection, which has been shown to further facilitate aptamer binding to saccharides at physiological pH[35]. We used a variety of experimental methods to characterize the various components and conjugation steps utilized for click-PD, including the two-step boronic acid modification of the DNA, the alkyne beads utilized for pre-enrichment, and the fluorophore-labeled glucose utilized for FACS screening. These results are detailed in Supplementary Figs. 10–16.

After three rounds of click-PD screening (Supplementary Fig. 17), we identified five candidate sequences with the highest copy number in the final-round sequencing data. Of these, we determined the sequence that had the highest affinity for glucose (NNG; Supplementary Table 3) and sought to identify the minimal binding region, as the full-length sequence contains twenty modified bases and would complicate solid phase synthesis of the aptamer strand for the screen (Supplementary Fig. 18). Additionally, it has been shown that the full-length sequence is often not necessary to retain molecular recognition capabilities[36]. Secondary structure analysis was used to guide truncation of NNG, and this revealed that the primary structural element was a minimized 30-nucleotide hairpin near the 3' terminus of the sequence with only eight modified bases, which we hypothesized to be the binding site (Fig. 4a). This stem-loop structure appeared to be conserved in three of the five candidate sequences, further supporting the importance of this region. We generated beads displaying this truncated sequence, NNGmin, and compared its binding performance relative to NNG using flow cytometry. NNG exhibited a $K_D$ of 1.9 mM—approximately an order of magnitude better than the best natural DNA

aptamer to glucose[26]—and we observed a small further increase for NNGmin in affinity in comparison to the full-length sequence (1.1 mM) (Supplementary Fig. 19). These results demonstrated that the minimized NNGmin sequence would be suitable for developing a glucose-responsive switch. To confirm the successful conjugation of the boronic acid moieties to the minimized aptamer, we performed analysis via electrospray ionization tandem mass spectrometry (ESI-MS). Surprisingly, although the starting aptamer (Bio-NNGmin) and DBCO-conjugated intermediate (BioNNGmin+DBCO) showed the expected masses (Supplementary Figs. 20, 21), the final boronic acid-conjugated product (BioNNGmin+BA) had only seven of the eight expected boronic acid modifications (Supplementary Fig. 22). Our hypothesis is that this is a consequence of steric hindrance induced by the secondary structure of the folded aptamer, which prevented one of the DBCO groups from reacting with the boronic acid.

Using our ADS screen, we were able to identify non-natural aptamer switches which were responsive to glucose concentrations in the low millimolar range. Although our aptamer is base-modified, we could still utilize the same natural DNA switching strand library, highlighting the generalizability of our strategy. We screened our ADS library against two concentrations of glucose, imaging with four alternating cycles each of buffer, 10 mM glucose, and 100 mM glucose. Interestingly, while we identified a few weakly-performing signal-on switches, glu1min primarily yielded signal-off switches that exhibited a decrease in fluorescence upon binding glucose. We cannot a priori predict the mechanism by which the switches identified in our screen will function, and the amplitude and direction of the fluorescence response will be dictated by factors such as the nature of the interaction between SD region and aptamer, steric clashing between the switching and aptamer strands, and the three-dimensional structure of the aptamer. Analysis of the filtered data revealed that 200 of the ~495,000 clusters exhibited a 1.3-fold change in fluorescence intensity upon the addition of glucose, with the most responsive sequence showing a 2-fold change (Supplementary Fig. 23). Although the magnitude of the signal change for the top sequences was modest compared to the ATP screen (approximately 50% lower), we were encouraged to observe similar decreases in fluorescence with both 10 and 100 mM glucose, suggesting that the top switching domains remained highly target responsive in the low millimolar range.

To gain insight into which bases within the aptamer were key for switching function, we subsequently performed MEME analysis of the top 1000 SD sequences to identify any conserved motifs (Supplementary Fig. 24). We found that bases 2–5 near the 5′ terminus of glu1min were overwhelmingly targeted by 15.7% of the top sequences (Fig. 4a). A histogram of the frequency with which each base-position in the glucose aptamer was complementary to one of these top-performing SD sequences also showed a prominent peak at the 5′ end, in keeping with the MEME analysis (Fig. 4b). Currently, there are no software tools that can accurately model the folding patterns of base-modified nucleic acids. However, we posited that modeling them using conventional DNA folding software may yield helpful patterns in terms of motif analysis. Interesting, it showed that targeting of bases 2-5 by the m1 motif maybe highly amenable to strand-displacement-based switching where target binding favors the formation of the intramolecular stem region and release of the SD sequence. We also noted that the other predicted short stem region between bases 9–12 appears to be the least amenable region of the aptamer for switch design. We hypothesize that this difference between the two stem regions arises because three of the four base pairs in this predicted stem from bases 9-12 involve a modified base, which could potentially destabilize this stem and even completely prevent it from forming. Although the boronic acid modification is incorporated at a position facing away from the binding face of the uracil base, the presence of modifications

nevertheless results in steric hindrance that has been shown to lower the stability of nucleic acid duplexes[37]. We emphasize that these explanations are highly speculative, given the limitations of computational modeling of base-modified nucleic acids. However, it highlights the unique obstacles in utilizing non-natural nucleic acids for switch development, which we overcome during the screening process as no prior knowledge of the aptamer folding is required.

As with the ATP ADS constructs, sequence alignment analysis revealed that the SD sequences exhibited notably low complementarity with the non-natural glucose aptamer. Approximately 73% of the top 1000 sequences showed a Smith-Waterman score ≤ 4 to the reverse-complement of the aptamer (Fig. 4c), demonstrating that more than half of the bases within top-performing SD sequences would not be expected to interact with the aptamer through canonical Watson-Crick base-pairing. We hypothesize that the grouping of boronic acid modifications in bases 5–8 and 9–12 likely contributes to the low SD complementarity due to the destabilizing effect of the base modification, which results in alternative interactions which cannot be predicted or modeled with current tools. Thus, our analysis again shows that many of the best-performing SD sequences are largely composed of bases that participate in non-canonical binding modes that would most likely not be identified in a rational design-based effort.

Finally, we identified and experimentally validated four NNGmin-based switch constructs with low millimolar affinity for glucose. We chose four switches (glu1–4; Supplementary Table 4) that exhibited the greatest signal-off response (1.3–2-fold decrease) in the presence of target, which we validated via cluster image analysis (Fig. 5a, b). We then assessed the glucose affinity of these four sequences in a plate reader assay, as described for the ATP screen above. All four sequences displayed the expected dose-dependent decrease in fluorescence, with a maximum signal change of approximately two-fold—comparable to that observed in the flow-cell. All four ADS constructs possessed affinities on the order of ~0.3 mM, making these constructs ~30-fold more sensitive to glucose than a previously published natural DNA-based switch[26] (Fig. 5c, Supplementary Fig. 25). We were surprised to observe that our ADS constructs possessed affinity that was comparable and even slightly superior to that of both the original NNG aptamer and the NNGmin derivative. In order to assess the specificity of these ADS constructs, we also tested various control versions of the glu1–4 constructs. Specifically, these included constructs incorporating the natural DNA analog of NNGmin, a scrambled version of NNGmin, and scrambled SD sequences. While we observed modest glucose response from some of these control sequences, the signal change was less (7–10% for natural DNA and scrambled SD; 20% for scrambled NNGmin) compared to the original glu1-4 switches (45-50% signal change) (Supplementary Fig. 26A–C). We hypothesize that the higher non-specific activity of the scrambled NNGmin results from the presence of the eight boronic acid modifications, which encourages folding patterns that are conducive to glucose binding even when the locations are altered from the original sequence. This is not surprising, as the strong interaction of glucose and boronic acid will likely perturb interactions between the two strands. However, the approximately 2-fold decrease in signal change that results from scrambling the modifications indicates that the original locations of the boronic acids are necessary for optimal folding response. We further explored the selectivity of our ADS switches by investigating binding to an alternative sugar, fructose (Supplementary Fig. 26D). We found that the ADS switches bound fructose and glucose with similar affinities. The significant off-target binding to fructose that we observed is not surprising, due to its close structural similarity to glucose and the fact that boronic acids can interact with any saccharide diol. It is likely that

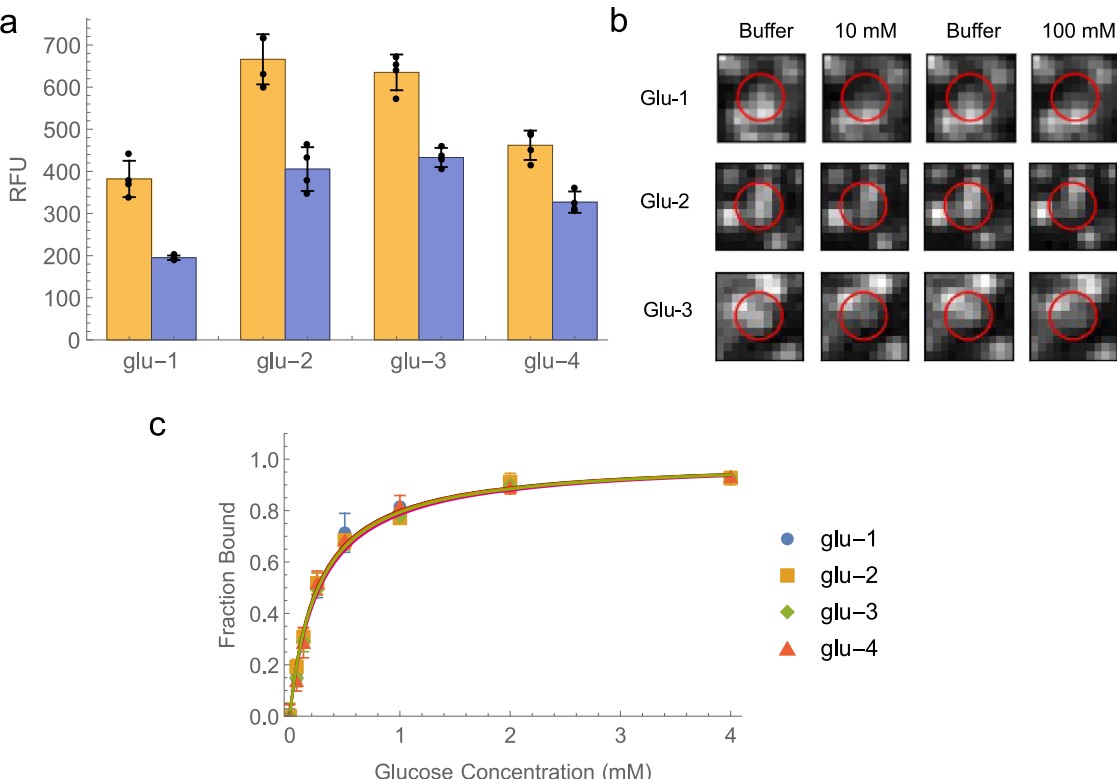

**Fig. 5 | Identification and characterization of phenylboronic acid-modified aptamer switches for glucose. a** Analysis of the top four aptamer switch clusters identified in our flow-cell screen. Orange and blue bars represent cluster intensity in buffer and 10 mM glucose, respectively. Data are presented as mean values ($n = 4$ replicates) and error bars represent the standard deviation of the measurements. Individual data points are overlayed as black dots. Source data are provided as a Source Data file. **b** Extracted images of clusters glu-1, −2, and −3 (red circles) on the MiSeq flow cell for both the buffer and glucose cycles. **c** Validation of the glucose affinity of the four aptamers shown in panel A. 50 nM labeled ADS construct was incubated with various concentrations of glucose and the fluorescence signal was measured on a plate reader. The solid lines represent the fitted single binding site model. Data are presented as mean values ($n = 3$ replicates) and error bars represent the standard deviation of the measurements. Source data are provided as a Source Data file.

the use of a counterselection step against fructose during the selection process could have increased aptamer specificity towards glucose.

Traditionally, the conversion of an aptamer to a strand displacement-based switch results in an increase to the apparent $K_D$, as higher target concentrations are needed to outcompete the complementary strand. These results highlight that the non-canonical interactions identified using our screening method could prove advantageous in the identification of ADS switch constructs which retain the affinity of their parent aptamer, and show that base-modified DNA can be advantageous for the development of high-affinity switches for challenging small-molecule targets.

In this work, we describe a massively parallel screening strategy for generating target-responsive molecular switches from an existing aptamer in a single experiment. Importantly, our strategy does not require any prior knowledge of aptamer structure and eliminates the need for computer modeling or design heuristics. To demonstrate the generalizability of our method, we converted two aptamers—a natural DNA aptamer for ATP, and a newly-selected phenylboronic acid-modified DNA aptamer for glucose—into molecular switches. Many of the resultant aptamer switches were either predicted not to fold or yielded inconsistent results when analyzed with various secondary-structure prediction software tools, which indicates that these switch constructs would have been overlooked by rational design-based approaches. Indeed, several of the aptamer sequence elements preferentially enriched for in our screen would have been counter-intuitive based on conventional design heuristics—for example, targeting one of the loop structures in the ATP aptamer, or favoring one predicted stem versus another

in the glucose aptamer. We have also demonstrated that our screen is applicable to base-modified aptamers for which in silico prediction of aptamer structure may not be feasible. Since our approach is predicated on the use of SD libraries composed entirely of natural DNA, this same system should be applicable to RNA aptamers or other non-natural aptamer chemistries such as xeno nucleic acids (XNA)[38], as long as the aptamer molecule is capable of stable hybridization with a DNA-based switching strand. Finally, while only a single switching strand design was used in this study (consisting of an N10 SD and 25-nucleotide poly-T linker), other switching strand libraries could be utilized in the aptamer switch screening platform, and future work could explore the optimal design of such switching strand libraries.

Even as the number of published aptamers steadily grows, it has remained a challenging and time-consuming task to convert those aptamers into functional molecular switches. This work addresses this fundamental challenge by relying on unbiased high-throughput screening to cover the full sequence space available for an ADS construct, rather than relying on rational design informed by potentially faulty assumptions of aptamer structure and function. While the switch screening platform does require the purchase and minimal modification of a relatively costly Illumina MiSeq sequencer, the screen itself is very rapid (~48 hours) and costs nearly the same as a traditional sequencing run, where the majority of the cost is the sequencing kit. And whereas conventional methods can only screen and validate a handful of aptamer switches at a time at a similar price-point, our screening technology can evaluate millions of aptamer switches in a single experiment. Our approach should be generalizable to many aptamers and should therefore

help accelerate the generation of functional aptamer-based switches, thereby facilitating the creation of biosensors for use in a broad range of applications.

# Methods

## Materials

Terminal transferase (TdT) was ordered from New England Biolabs (#M0315L). Ddel restriction enzyme was ordered from New England Biolabs (#R0175L). Cy3-labeled ddUTP (5-propargylamino-ddUTP-Cy3) was ordered from Jena Bioscience (#NU-1619-CY3) and unlabeled ddTTP (2′,3′-dideoxythymidine-5′-triphosphate) was ordered from TriLink Biotechnologies (#N-4004). ATP (adenosine 5′-triphosphate) was ordered from Thermo Fisher Scientific (#R0441) and glucose was purchased from Sigma-Aldrich (#G8270). The strands for the switch screen were all purchased HPLC-purified from Integrated DNA Technologies (IDT); all purchased sequences are presented in Supplementary Tables 1 and 3. The switching domains that were validated via plate-reader were purchased from the Stanford Protein and Nucleic Acid facility and the sequences are presented in Supplementary Table 2. Streptavidin beads for the proof-of-concept TdT and Ddel experiments were purchased from Thermo Fisher Scientific (#88816). All other reagents were purchased from Sigma-Aldrich unless otherwise noted.

## Conjugation of glucose to magnetic beads

Glucose was immobilized onto alkyne-coated magnetic beads (Jena Bioscience) using CuAAC click chemistry. 400 μL of beads were washed three times with 400 μL of 0.1% Tween-20 in PBS. The washed beads were resuspended in 1X PBS, 30 μL of a pre-prepared mixture of 0.1 M CuSO4/0.2 M Tris(3-hydroxypropyltriazolylmethyl)amine (THPTA), 30 mM azido-PEG4-β-D-glucose (BroadPharm), and H$_2$O to a final volume of 225 μL. 25 μL of freshly-prepared 50 mM sodium ascorbate was added to the reaction mixture for a final concentration of 5 mM. The solution was degassed using N$_2$ for 5 minutes, then reacted for 1 hr at room temperature with rotation. Beads were then washed three times and resuspended with 400 μL 1X SELEX buffer (100 mM NaCl, 2 mM MgCl$_2$, 5 mM KCl, 1 mM CaCl$_2$, 0.02% Tween 20, 20 mM Tris-HCl, pH 7.5). The same protocol was utilized to conjugate Alexa Fluor 488 azide to the alkyne beads for validation of the click chemistry efficiency.

## Conjugation of AlexaFluor 647 to magnetic beads

Alexa Fluor 647 was immobilized onto Dynabeads M-270 amine magnetic beads (Thermo Fisher Scientific) using standard amine reactive chemistry. 400 μL of amine beads were washed three times with 400 μL 0.1% Tween-20 in PBS. 1 mg Alexa Fluor 647 NHS ester (Thermo Fisher Scientific) was resuspended in dimethylformamide (DMF) to a final concentration of 10 μg/μL. The washed beads were resuspended in a 400 μL solution comprising 24 μL Alexa Fluor 647 NHS-ester stock in 1X PBS. The bead mixture was incubated for 2 hrs at room temperature (RT) with rotation. Beads were then washed three times and resuspended with 400 μL 1X SELEX buffer.

## Characterization of small molecules and oligonucleotides

All mass spectra were obtained in negative mode. High resolution mass spectra (HRMS) of Alexa Fluor 647 alkyne, Sulfo Cy5 alkyne (Lumiprobe) and Alexa Fluor 647 conjugated glucose were obtained by Stanford University's Vincent Coates Foundation Mass Spectrometry Lab on a Waters Acquity UPLC and Thermo Exploris 240 Orbitrap mass spectrometer equipped with an Agilent Zorbax SB-C18 column (2.1 × 50 mm, 80 Å, 2.7 μm). Spectra were collected in full scan MS mode with polarity switching (collecting scans alternating between positive and negative ionization potentials), Orbitrap resolution 120000, mass range of 150-2000 Daltons with RunStart Easy-IC internal mass calibration. The gradient is provided in Supplementary Table 5. HPLC

chromatograms of Alexa Fluor 647 alkyne, and Sulfo Cy5 alkyne were obtained on an Agilent 1290 Infinity II UHPLC equipped with a Waters XBridge Oligonucleotide BEH C18 Column (2.1 × 50 mm, 130 Å, 2.5 μm). The gradient is provided in Supplementary Table 6. The ESI-MS analysis of oligonucleotides was carried out by Novatia, LLC (Newtown, PA).

## Pre-enrichment by SELEX for glucose

Two rounds of positive selection were performed with bead-immobilized glucose. All bead washing steps were performed using a Dynamag-2 magnetic separation stand (Thermo Fisher Scientific). 1 nmol of the GluPD library (Supplementary Table 3) was folded in 200 μL of 1X SELEX buffer by heating to 95 °C for 5 min, cooling at 4 °C for 10 min, and then incubating at 25 °C for 10 min. In the first round of pre-enrichment, we included a counterselection step where the folded library was incubated with 5 nmol (20 μl) of Alexa Fluor 647-conjugated magnetic beads for 1 hr at RT with rotation to eliminate sequences that preferentially bind the fluorophore. These beads were then washed twice with 100 μl 1X glucose selection buffer, and the total 200 μl of buffer that was washed from the Alexa Fluor 647 beads were then incubated with 5 nmol (20 μl) of glucose-conjugated magnetic beads which were pre-washed as described above for the Alexa Fluor 647 beads for 1 hr at RT with rotation. Beads were washed twice with 100 μl 1X SELEX buffer and eluted into 100 μl water by heating to 95 °C for 5 min. Recovered DNA was purified with a Qiagen MiniElute cleanup kit and eluted in 10 μl water. We then PCR amplified 1.5 μl of the recovered library with 10 μl 2X GoTaq PCR mix, 200 nM Glu FP, 200 nM biotinylated Glu RP, and H$_2$O up to a final reaction volume of 50 μl using the following cycling conditions: 95 °C for 3 min, followed by X cycles of 95 °C for 15 s, 53 °C for 30 s, 72 °C for 30 s and finally 72 °C for 2 min. To determine the correct number of cycles for amplification, a pilot PCR was run. 3 μl of the reaction was removed every 3 cycles (15 cycles total), mixed with 3 μl 2X Novex TBE-urea sample buffer (Thermo Fisher Scientific) and run on a 10% TBE gel at 200 V for 40 min. The cycle that yielded a product of the correct length without forming undesired products was chosen for the final scaled-up PCR reaction. To generate single-stranded DNA, biotinylated double-stranded DNA was immobilized onto 100 μl MyOne streptavidin C1 magnetic beads (Thermo Fisher Scientific) according to the manufacturer's protocol in 500 μl 1X Binding and Washing buffer (5 mM Tris-HCl (pH 7.5), 0.5 mM EDTA, 1 M NaCl) in a 1.5 mL tube. Beads were incubated with 100 μl freshly prepared 0.5 M NaOH for 10 minutes at RT. The tube was placed on a DynaMag-2 magnetic rack (Life Technologies) for 2 minutes, and the supernatant was collected. Beads were washed once more with 50 μl 0.1 M NaOH. DNA was recovered from NaOH by adjusting the pH with 25 μl 3 M NaOAc, then purified with a Qiagen MiniElute cleanup kit and eluted in 20 μl water.

## Forward primer (FP) bead conjugation protocol

500 μL of Dynabeads MyOne carboxylic acid magnetic beads (Thermo Fisher Scientific) were washed five times with 500 μL of water on a magnetic rack. The beads were then resuspended in 150 μL of 0.2 mM 5′ amino-PEG Glu FP, 200 mM NaCl, 1 mM imidazole chloride and 250 mM EDC. The mixture was mixed well and sonicated prior to incubation at RT overnight on a rotator. Next, we conjugated PEG12 to the unreacted free carboxyls on the magnetic beads through a two-step NHS/EDC reaction to reduce non-specific interaction with the target proteins. The beads were washed three times with 500 μL of 100 mM MES buffer (pH 4.7). During the last wash step, the beads were incubated for 10 minutes at RT on a rotator. Immediately before use, an 80 mg/mL solution of EDC and a 25 mg/mL solution of NHS were prepared in cold 100 mM MES buffer. The FP beads were then resuspended in equal volumes of NHS and EDC solutions to a final volume of 150 μL. The beads were mixed well and incubated at RT on a rotator for 30 minutes. The beads were washed twice with 500 μL of cold PBS. The

activated beads were then resuspended in 150 µl of 20 mM amino-PEG in PBS, mixed well, and incubated for at least 30 minutes at RT on a rotator. The beads were then washed three times for 15 minutes with 500 µL of 1X SELEX buffer in order to quench any amine-reactive NHS esters. Finally, the beads were resuspended in 500 µL of 1X SELEX buffer and stored at 4 °C. To verify successful attachment of the primer to the beads, 1 µl of FP beads and 1 µL of 100 µM Alexa Fluor 647-labeled FP complement were mixed in 100 µl SELEX buffer and incubated for 10 minutes at RT on a rotator. The beads were then washed once with 100 µL of 1X SELEX buffer, resuspended in 100 µL 1X SELEX buffer, and run on a benchtop flow cytometer (BD Accuri C6 Plus).

### Emulsion PCR protocol
The emulsion PCR process involves the creation of an oil phase and an aqueous phase. The oil phase consists of 4.5% Span-80, 0.4% Tween 80, and 0.05% Triton X-100 in mineral oil (all purchased from Sigma-Aldrich), stored at RT in the dark. The aqueous phase consists of 1X KOD XL buffer, 0.5 U of KOD XL polymerase, 0.2 mM dATP, 0.2 mM dCTP, 0.2 mM dGTP, 0.2 mM aminoallyl dUTP (all purchased from Thermo Fisher Scientific), 10 nM FP, 1 µM RP, 1.5 pM dsDNA enriched glucose library, and ~3 × 10$^8$ FP-coated magnetic beads (12 µL of FP-bead suspension) in a total volume of 1 mL of water. To create the water-in-oil emulsions, 7 mL of the oil phase was added to a DT-20 tube (IKA) and 1 mL of the aqueous phase was added dropwise over ~30 seconds while the mixture was stirred at 620 rpm in an Ultra-Turrax device (IKA). The mixture was then stirred on the device for another 5 min. The emulsion was hand pipetted into ~80 wells of a 96-well PCR plate (100 µL per well), which was run on an Eppendorf Mastercycler X50 PCR machine for 40 cycles.

### Emulsion cleanup
After PCR, the emulsions were transferred to a 50 mL Falcon tube. 125 µL of 2-butanol (Thermo Fisher Scientific) was added to each well to wash residual emulsion, and the butanol was then transferred to the same 50 mL tube. The tube was vortexed for 30 seconds and then centrifuged at 3000 × g for 5 minutes. The supernatant was removed while retaining the pellet of aptamer particles at the bottom of the tube. 1.2 mL of breaking buffer (100 mM NaCl, 1% Triton X-100, 10 mM Tris-HCl, pH 7.5, and 1 mM EDTA) was added to resuspend the particles, and the mixture was transferred to a new 1.5 mL tube. The 1.5 mL tube was vortexed and centrifuged at 21,000 x g for 1 minute. Using a magnetic rack, the supernatant was removed with a 1 mL micropipette. Another 1 mL of breaking buffer was added to the particles, which were then transferred to a new tube, and the supernatant was removed as described above for multiple cycles until no residual oil (white film) was visible on the top layer. On the last wash, 150 µL of breaking buffer was added to the sample, which was then transferred to a new 1.5 mL tube. The supernatant was removed, and the aptamer particles were washed three times and resuspended with 500 µL SELEX buffer.

### Boronic acid functionalization of aptamer particles
Aptamer particles recovered from the emulsion PCR were washed twice with 200 µL 1X PBS and resuspended in a 150 µL solution containing 1X PBS, and 30 µL of a 10 mg/mL solution of DBCO NHS ester. The beads were incubated for 2 hrs at RT with rotation. Beads were washed three times and resuspended in 0.1% Tween-20 in PBS. The DBCO-modified aptamer particles were washed three times with 200 µL 0.1% Tween-20 in PBS. The washed beads were resuspended in a 150 µL solution containing 1X PBS and 40 µL of a 1 mg/mL solution of azido-phenylboronic acid. The beads were vortexed and incubated at RT overnight. The boronic acid-modified beads were then washed three times and resuspended with 100 µL 1X SELEX buffer.

### Single-stranded DNA generation and aptamer particle quality control
The aptamer particles were resuspended in 500 µL of 200 mM NaOH and incubated for 10 min at RT on a rotator. The aptamer particles were washed twice with 500 µL of 100 mM NaOH and then three times with 1 mL of 1X SELEX buffer and finally resuspended in 100 µL of 1X SELEX buffer. To ensure the successful synthesis and monoclonality of the aptamer particles, 1 µL of the aptamer particle solution and 1 µL of 100 µM Alex Fluor 647 Glu RP were incubated in a total volume of 100 µl 1X SELEX buffer for 10 minutes at RT with rotation. The beads were washed once with 100 µl of 1X SELEX buffer, resuspended in 50 µl 1X SELEX buffer, and run on a flow cytometer (BD Accuri C6 Plus).

### Glucose labeling
Glucose was conjugated to Alexa Fluor 647 or Alexa Fluor 488 using CuAAC click chemistry. Solutions were prepared containing 1X PBS, 30 µL of a pre-prepared mixture of 0.1 M CuSO$_4$/0.2 M tris(3-hydroxypropyltriazolylmethyl)amine (THPTA), 30 mM azido-PEG4-β-D-glucose (BroadPharm), 50 µL of a 10 mg/ml stock of either Alexa Fluor 647 alkyne or Alexa Fluor 488 alkyne (Invitrogen), and H$_2$O to a final volume of 225 µL. 25 µL of a freshly-prepared 50 mM solution of sodium ascorbate was added to the reaction mixture for a final concentration of 5 mM. The solution was degassed using N$_2$ for 5 minutes, then reacted for 1 hr at room temperature with rotation. After the labeling reaction was complete, the samples were dialyzed in water using Tube-O-DIALYZER Micro 1 K MWCO (G-Biosciences).

### Click-PD screening for glucose-specific aptamers
Three rounds of particle display were performed with Alexa Fluor 647-labeled glucose, and one final round was performed with Alexa Fluor 488-labeled glucose to eliminate any sequences that may have evolved to preferentially bind the fluorophore. Prior to sorting, a flow cytometry binding assay was performed with multiple concentrations of glucose (250, 500, 750, 1000 µM) to determine which resulted in sufficient binding to the target. For the assay, 1 µL of the boronic acid-modified aptamer particle solution was added to a solution containing labeled glucose at the specified final concentrations in a total reaction volume of 50 µL of 1X SELEX buffer and incubated in the dark for 1 hr at RT on a rotator. Samples were then washed with 100 µL 1X SELEX buffer and resuspended in cold 1X SELEX buffer for analysis. The optimal concentration of glucose for sorting was determined as the highest concentration that yielded >1% binding to labeled glucose. For sorting, aptamer particles were folded in 1 mL SELEX buffer by heating to 95 °C for 5 min and leaving to cool to room temperature for 30 min. and then incubated with 750 µM labeled glucose in the dark for 1 h at room temperature with rotation. The beads were washed twice and resuspended in 1 mL cold 1X SELEX buffer and then analyzed using a BD FACS Aria III equipped with BD FACSDiva software (version 8.0.2). The sort gate was set to collect 0.5% (round 1) or 0.3% (rounds 2 and 3) of aptamer particles that showed high affinity for glucose by identifying those particles with the greatest shift in the APC channel (rounds 1 and 2) or FITC channel (round 3). After sorting, the collected aptamer particles were resuspended in 20 µL PBS and the aptamers were amplified by PCR using the conditions described above.

### High-throughput sequencing protocol
For each aptamer pool sequenced, adaptor primers were first added. 10 ng of double-stranded DNA was subjected to eight cycles of PCR using the same conditions described above. A 2x GoTaq Master Mix was used (Promega) with 1 µM of each primer in a total reaction volume of 100 µL. The sequencing primers were added by using a Nextera XT kit (Illumina) and following the provided instructions. Samples were quantified using a Qubit fluorometer and sent to the

Stanford Functional Genomics Facility for sequencing on an Illumina MiSeq.

## Synthesis of aptamer particles for affinity measurements

Aptamers were coated onto beads by preparing a 100 μL PCR reaction consisting of 10 μL 10X KOD XL buffer, 2 μL dNTP mix of 10 mM dATP, dGTP, dCTP, aminoallyl dUTP each, 1 μL of 10 μM Glu FP, 10 μL of 10 μM Glu RP, 2 μL of 100 pM aptamer template, 2 μL of KOD XL polymerase, and water to the final volume. 30 PCR cycles were conducted using the conditions described above, and the beads were washed and converted to single-stranded DNA as described above for the emulsion PCR protocol. Beads were washed and resuspended in 50 μL of SELEX buffer prior to storage at 4 °C. A 50 μL binding reaction was prepared with 1 μL of the aptamer particle solution and the required volume of the Alexa Fluor 647-labeled glucose stock in 1X SELEX buffer. The samples were incubated on a rotator at RT for 1 hr. The beads were washed twice with 100 μL cold SELEX buffer and resuspended in 50 μL SELEX buffer. The sample was gently mixed via pipette, sonicated briefly, and then immediately run on a flow cytometer (BD Accuri C6 Plus equipped with BD Accuri C6 plus software version 1.0.23.1).

## Preparation of the switching strand library

Prior to running the high-throughput switch screen, the switching strand library must be prepared for MiSeq sequencing. This process only needs to be done once, as the prepared library should be sufficient for many (~50) runs and is not dependent on the aptamer being used. To prepare the library for sequencing, a Nextera XT DNA Library Preparation Kit was used (Illumina) and the kit instructions were followed. Kit indices N703 and S517 were used, and the final double-stranded PCR product was checked via native PAGE and quantified using a Qubit fluorimeter prior to sequencing.

## Validation of TdT and DdeI enzymes on beads

Prior to MiSeq screening, the TdT labeling reaction and enzymatic cleavage reaction using DdeI were validated using streptavidin-functionalized magnetic beads and a bench-top flow cytometer (BD Accuri C6 Plus Flow Cytometer).

To validate the DdeI restriction enzyme, we utilized a 5′-biotinylated test strand (see Supplementary Table 1) that was 3′-labeled with Cy3. The test strand contained the same poly-T linker, N10 SD region, and reverse-primer complement region (which contains the DdeI cut site) as the switching strand library that was utilized in the screen. The test strand was captured onto streptavidin beads and we then annealed the reverse-primer at 100 nM, washed the beads, and incubated the beads with 3 μL DdeI enzyme, 10 μL 10X Cutsmart buffer, and 87 μL water for ten minutes at 37 °C. By comparing the fluorescence of the beads before and after the addition of the restriction enzyme, we were able to confirm that DdeI efficiently cleaved the reverse-primer complement region from the aptamer.

To validate the TdT labeling step, we checked the incorporation of a Cy5-labeled ddUTP into the 20-nucleotide biotinylated reverse-primer sequence. The primer sequence was subjected to a test TdT reaction with 5 μL 10X TT buffer, 1 μl 10 μM biotin RP, 5 μL CoCl₂, 15 μL 1 mM Cy5 ddUTP, 5 μL TdT enzyme (all reagents from New England Biolabs), and 29 μL water and incubated at 37 °C for 0, 30, 60, or 90 minutes. Samples from these various time-points were then captured on streptavidin magnetic beads to analyze reaction efficiency, which could be assessed by a shift in the red-channel fluorescence corresponding to successful incorporation of Cy5-ddUTP. Analysis of these samples by flow cytometry showed that the reaction proceeded to completion quickly, with a large shift in fluorescence observed after 30 minutes and no significant changes after longer incubation times.

## High-throughput ADS screening

The high-throughput ADS screen was conducted using the custom-built N2A2 system we described previously[24]. The system uses a modified Illumina MiSeq instrument utilizing MiSeq control software version 2.6.2.1. The DdeI enzyme solution (10 μL DdeI enzyme stock, 30 μL 10X Cutsmart buffer, 260 μL water), complement strand solution (3 μL of 100 μM switch complement strand, 40 μL 10X Cutsmart buffer, 356 μL water), blocking TdT solution (30 μL TT buffer, 40 μL CoCl₂ buffer, 45 μL 2 mM dTTP, 16 μL TdT enzyme, 259 μL water), and Cy3-TdT labeling solution (30 μL TT buffer, 40 μL CoCl₂ buffer, 45 μL 1 mM Cy3-ddUTP, 16 μL TdT enzyme, 259 μL water) were all added into empty locations on the MiSeq reagent cartridge. Water, 50 mM NaOH with 1% SDS, selection buffer (20 mM Tris-HCl, 120 mM NaCl, 5 mM KCl, 1 mM MgCl₂, 1 mM CaCl₂, and 0.01% Tween-20 in nuclease-free water), FM buffer (100 nM FM comp 532 and 100 nM FM comp 660 in selection buffer), aptamer solution (100 nM aptamer in FM buffer), and target solutions (ATP or glucose in selection buffer) were all hooked up to the external multiport valve (Valvo Instruments).

The MiSeq XML files and folder agent version 2016 (123) were altered to conduct three different types of cycles: a switch construction cycle, a buffer cycle, and a target addition cycle. Unless otherwise mentioned, all steps were conducted at 22 °C. In the switch construction cycle, the flow-cell is first blocked with ddUTP by flowing in the TdT blocking solution for 45 minutes at 37 °C. This is repeated once more. The flow-cell is then washed with the NaOH solution, and then selection buffer. The complement strand solution is then injected onto the flow-cell and allowed to incubate for 15 minutes. Next, the DdeI solution is injected and allowed to incubate for 30 minutes at 37 °C. These two steps (complement strand and DdeI incubation) are repeated once again. The flow-cell is once again washed with NaOH solution and selection buffer prior to the final Cy3 labeling of the switching strand library. In this step, the labeling TdT solution is added to the flow-cell for 45 minutes at 37 °C. The step is repeated once more, and the switch construction cycle is completed after washing the flow-cell with NaOH solution and then selection buffer.

The buffer cycle involves annealing the quencher-labeled aptamer onto the switching library clusters and then imaging the flow-cell in buffer. The first step of the buffer cycle is to anneal the quencher-labeled aptamer. The aptamer solution is injected onto the flow-cell and then the flow-cell undergoes a temperature anneal (80 °C for 7.5 minutes, 70 °C for 2.5 minutes, 60 °C for 2.5 minutes, 50 °C for 2.5 minutes, 40 °C for 2.5 minutes, 30 °C for 2.5 minutes, 22 °C for 15 minutes). The flow cell is then washed with selection buffer and the clusters are imaged to determine the switch cluster intensities in the present of buffer without target.

In the target addition cycle, the target solution is injected onto the flow cell and then incubated for 5 minutes. This is repeated twice more for a total incubation time of 15 minutes between the target solution and the switch clusters on the surface of the flow cell. The flow-cell is then imaged to determine the switch cluster intensities in the presence of target. Prior to beginning the next buffer cycle, the flow-cell is washed with the NaOH solution and then selection buffer to remove the target and aptamer strands from the flow-cell.

## Evaluation of switches via plate-reader

For the plate-reader assay of the ATP and glucose aptamer switch constructs, we first annealed the Cy3-labeled switching strands to the quencher-labeled aptamer. The two strands were mixed together at a 1:1 ratio at a final concentration of 50 nM in 1x selection buffer. The solution was heated to 95 °C for 5 minutes and then cooled at RT for 20 minutes. For glucose switches, strands were annealed using a slow cooling from 95–25 °C by decreasing the temperature 2 °C every 30 s. 200 μL of the annealed switch constructs were then incubated for

 

10 minutes at RT with various concentrations of ATP or glucose in a black 96-well plate (Sigma-Aldritch). The measurements were taken at 25 °C on a Synergy H1 microplate reader (BioTeK, Gen5 software version 3.04.17) using a Cy3 filter cube (545 nm excitation, 575 nm emission). The experiment was conducted in quadruplicate, where each well corresponded to an independent incubation of the aptamer switch with target.

### Curve fitting and normalization

In order to normalize the binding curves and to determine the observed dissociation constants for the molecular switches, Wolfram Mathematica (Version 12.0.0.0) was used to fit binding models to the plate-reader data using the NonlinearModelFit function. Since the ATP aptamer has two binding sites, a two-site independent binding model was used:

$$f_r = \frac{K_1[\text{ATP}] + K_1 K_2 [\text{ATP}]^2}{1 + K_1[\text{ATP}] + K_1 K_2 [\text{ATP}]^2} \tag{1}$$

and adapted for the fluorescence measurements taken on the plate-reader:

$$y = (B_{\max} - y_0) \frac{K_1[\text{ATP}] + K_1 K_2 [\text{ATP}]^2}{1 + K_1[\text{ATP}] + K_1 K_2 [\text{ATP}]^2} + y_0 \tag{2}$$

In the first equation, $K_1$ is the dissociation constant of the first ATP target binding, while $K_2$ represents the dissociation constant of the second ATP target binding to the aptamer switch, and $f_r$ is the fraction bound. In the second equation, $y$ is the relative fluorescence signal detected on the plate-reader, $B_{\max}$ is the maximum signal at 100% binding, and $y_0$ is the initial fluorescence of the aptamer switch construct in the absence of target.

For the glucose aptamer, a single-site binding model was utilized:

$$f_r = \frac{[\text{Glucose}]}{[\text{Glucose}] + K_D^{eff}} \tag{3}$$

and adapted for the fluorescence measurements taken on the plate-reader:

$$y = (B_{\max} - y_0) \frac{[\text{Glucose}]}{[\text{Glucose}] + K_D^{eff}} + y_0 \tag{4}$$

### Switch binding kinetics assays

To screen the kinetics of atp-4, atp-5, and atp-6, we used the Synergy H1 plate reader with an injector module. Briefly, we filled the well with the respective switching construct, then used the injector to spike in a small volume of a highly concentrated ATP solution, and continuously (interval 0.46 s) measured the intensity of the well for one minute. We created the switch constructs using the same protocol as above (see Section "Evaluation of switches via plate-reader"), but with a final concentration of 60 nM in 1x selection buffer. We also prepared six different dilutions of ATP (0, 60, 180, 300, 600, and 900 μM) in 1x selection buffer. For each measurement, we filled a well of a 96-well plate (Sigma-Aldrich #CLS3993) with 75 μL of the respective switch construct, and then spiked in 15 μL of the ATP solution for the measurement. After spiking in the ATP solution, the final construct concentration in each well was 50 nM, with final ATP concentrations of 0, 10, 30, 50, 100, and 150 μM. All measurements were taken at 27 °C on the same plate reader. Experiments were conducted in triplicate, where each well corresponded to an independent spike-in of an aptamer switch with ATP.

### Statistics and reproducibility

Mean values and standard deviations for the experiments were conducted using Wolfram Mathematica (Version 12.0.0.0). For the high-throughput screening experiments and characterization of the aptamer switch reagents a minimum of three replicate measurements were taken so that a standard deviation could be calculated. The number of times each experiment was replicated is indicated in the figure captions, and each attempt to replicate experimental results was successful. No statistical method was used to predetermine sample size. The experiments were not randomized and no data were excluded from the analyses. The investigators were not blinded to allocation during experiments and the outcome assessment.

### Linking MiSeq data

Sequencing and intensity data from the MiSeq were linked before further analysis using previously published code[24,25]. The data from the MiSeq is separated into sequencing, flow-cell position, and intensity files in the formats of FASTQ, locs, and cif, respectively. FASTQ information contains the sequence, tile, and x, y positions (as integers) in the corresponding tile for each cluster on the flow-cell. The locs files contain the x, y positions (as 32-bit floating point numbers) of all clusters in a tile. The cif files contain the intensities for each of the four channels for each cluster in a tile in a given cycle. Before linking, FASTQ files are parsed and separated by tiles to reduce the amount of data loaded at once. During data linking, files are loaded by tile and matched using the x and y coordinates. The location data from locs and intensity data from the cif files share indices and do not need further matching. The x and y positions from locs files are converted to the FASTQ format through a simple transformation, where the coordinate from the locs file is multiplied by 10, increased by 1000, and then rounded. To accommodate for possible differences in rounding reported in the FASTQ coordinates (e.g. 12345.6 is rounded to 12345), all permutations of floor and ceiling values are also tried during the coordinate matching. For the overall linking process, data is linked for one tile at a time. For the chosen tile, the loc file and cif files across all desired cycles are loaded and stored. Then, working through each FASTQ, the sequence and coordinate data are extracted and matched to the subset of coordinates in the loc file. The linked data are then filtered if desired and written out to the designated file as sequence, coordinate, and intensity.

### Reporting summary

Further information on research design is available in the Nature Portfolio Reporting Summary linked to this article.

## Data availability

The raw sequencing data generated during the aptamer switch screen experiments are available under NCBI Sequence Read Archive (SRA) (Accession code: PRJNA952942"). The underlying processed data files from the high-throughput screen for both the ATP and glucose N2A2 runs are provided as Supplementary Data 1 and 2. Source data are provided with this paper.

## Code availability

The MiSeq data linking code for the platform is available on our GitHub account (https://github.com/sohlab/non-natural-aptamer-array), and the corresponding pseudocode is provided in the Methods section of the manuscript.

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

## Acknowledgements

We thank Professor Eric V. Anslyn and Caroline M Hinson for providing the boronic acid modification used for the non-natural glucose aptamer selection. We would also like to thank Dr. John Coller for his assistance developing and maintaining the N2A2 platform. This work was supported by the Chan-Zuckerberg Biohub, W. L. Gore & Associates, the Helmsley Trust, Stanford Spectrum Medtech program, the Biomedical Advanced Research and Development Agency (grant 75A50119C00051), and the NIH (grants OT2OD025342 and R01GM129314-01) (H.T.S.). A.M.Y. was supported by the Stanford Bio-X Graduate Fellowship.

## Author contributions

A.M.Y. and H.T.S. conceived of the screening platform technology and designed the initial experiments. A.A.H. conceived of the underlying ADS switch architecture that was utilized in the project. A.M.Y. built and characterized the switch screening platform, as well as conducting the ATP switch screen, analyzing the ATP switch data, and characterizing the ATP switches. L.A.H. designed and conducted the ATP binding kinetics experiments. A.E.R. selected and characterized the base-modified glucose aptamer. A.E.R. and A.M.Y. conducted the glucose switch screen and analyzed the data. A.E.R. characterized the glucose switches. L.W. managed and

executed the software pipeline that was used to process the raw MiSeq data files. L.Z. conducted and analyzed the mass-spectrometry experiments. A.M.Y., A.E.R., M.E., and H.T.S. wrote the manuscript.

## Competing interests

The authors declare no competing interests.
