## [Peer Review File · Nature Communications]

REVIEWER COMMENTS

Reviewer #1 (Remarks to the Author):

In this study, Yoshikawa et al., reported an Illumina MiSeq-based high-throughput screening strategy to engineer aptamers into molecular sensors. This similar platform has been used by the authors previously to select base-modified aptamers (Ref 24) and to improve the specificity of the aptamers (bioRxiv 10.1101/2021.11.01.466780). Here, the authors further demonstrated that molecular switches can be identified by screening a library of random 10 nucleotide-long switch domain from binding with either an ATP aptamer or a glucose aptamer. The major advantage of this strategy, as compared to conventional rational design strategy, is its throughput and generalizability, as it does not require prior knowledge of aptamer structure, meanwhile, non-natural nucleotide-based aptamer can also be used for the screening. While this is an interesting approach, however, the requirement of a specialized (and expensive) MiSeq instrument can easily prevent the broad usage of this method in general laboratories. More importantly, the relatively small signal-to-noise ratio of these identified sensors (e.g., for glucose, maximally only 1.5-2-fold decrease in fluorescence) further reduce the reviewer's excitement on this approach. Some other comments are listed below.

1. For both ATP and glucose switches, the motif nearby the 5'-end was identified as the most abundant hits. The authors suggested that was surprised (P10), however, that was not surprised to the reviewer because the quencher was modified at this 5'-end of the aptamer, and it has been well known that the fluorescence quenching is highly distance-dependent. It raised the first concern, even though the random switch domain was used, while actually the real "binding site" in the aptamer may be already predetermined by the location of the quencher. Or at least, it will largely prefer that site. That may be also one of the reasons why only a small change in the fluorescence signal was observed, because this 5'-end may not be the ideal binding site to engineer a molecular switch. As in the case of ATP aptamer, it may be not close to the binding pocket.
2. Following this point, whether this approach can only be used for short-stranded aptamers, e.g., 27 or 30 nt-long strands used in this study? This is important because both natural riboswitches and many synthetic aptamers can be much longer.
3. Why did the authors choose to use a 10-nucleotide-long switch domain? How will altering the length of this domain affect the switching behavior?
4. While the authors indicated the SELEX process to identify glucose aptamers, the detailed sequencing result used to identify NNG and the process and results of shortening NNG to obtain NNGmin should be provided. Meanwhile, the target selectivity of this aptamer should be examined.
5. The chemical structures of those boronic acid-modified bases should be provided.
6. The authors should have a brief explanation of why choosing the current poly T structure and length and also its potential interactions with the aptamers.

7. In the introduction, the authors claimed that their switches “represent the first example of a base-modified aptamer switch”, which is clearly not true! Many studies have been using modified bases to engineer aptamer sensors, just not name it as “aptamer switch”, see just for example, Sci. Rep 2017, 42716 or Bioconjugate Chem. 2011, 22, 282-288.
8. P3 “engineered riboswitches are also utilized as signaling molecules”, the use of “signaling molecules” can be confusing because of its real biological meaning.
9. P5, the authors discussed that “The ‘switching strand’... is responsible for endowing the construct with the ability to undergo target binding-induced conformational switching”. The reviewer may not agree with this point. Even without the switching strand, likely these aptamers can still undergo target binding-induced conformational switching and changes.
10. In Page 6 second last sentence, it is also possible that conformational change itself is not enough to induce any change in the fluorescent signal, which should be mentioned.
11. Figure S2, why there was a decrease in the fluorescence intensity, because a fluorescently-labeled DNA strand was used? The current figure caption didn’t provide any such information.
12. In Figure S3, why the percentage of R3 region signal increased in the first 30 min, and then decreased?
13. As mentioned in my first point, since it was not a surprise that m1 motif can be abundant, the discussion in the P10, “and this indicates that the short DNA loop...” part should be revised.
14. As shown in Figure S7, the fold increase in the fluorescence signal was much reduced as compared to that on the flow cell. It should also be discussed in the main text. Why is that?
15. Similarly, while atp-7 exhibited the highest affinity, it also exhibited the lowest signal-to-noise ratio. It should be discussed.
16. “Monoclonal aptamer particles” is not a typical phrase, more like a jargon, which should be revised or explained.
17. The format of references 24, 28, 36, 39 should be revised.
18. Some typos: In Figure 1C, “High-throughput screening”; Figure S1, “Ddel”; In Figure 4C, label of y-axis is missing; “Thus, so we carried out the analysis, fully recognizing is limitations (see methods)”. Also, it is a bit confusing what the methods are referring to.

Reviewer #2 (Remarks to the Author):

Yoshikawa et al. describe a high-throughput approach to identifying novel switch aptamers. Switches are usually discovered in binding selections and then designed for activity through some combination of computational and rational design. These approaches are typically fraught because design rules are

incomplete or sometimes wrong. Here, an array of aptamers is generated on an illumina sequencing slide and interrogated for function by hybridizing a quencher strand and monitoring the array for changes in fluorescence after the target molecule is added. The approach is demonstrated for an ATP-sensing switch and a glucose-sensing switch. While the approach is potentially interesting, the data are largely speculative and none of the discovered molecules or reagents used are characterized to a level that supports the conclusions. Publication is premature at this time. I provide major concerns.

MAJOR CONCERNS

1. None of the reagents or discovered molecules are characterized. Multiple times, methods are presented that assume that click reactions yield only product. While these reactions are efficient, this is dramatic oversimplification and it could have profound implications for the screen. Characterize, for example, the glucose-dye conjugates, the alkyne beads, and the azido phenyl boronic acid products. Routine chemical analysis includes mass spectrometry, liquid chromatography, and/or NMR spectroscopy. Characterizing the boronic acid products will be more difficult. Consider a Suzuki cross-coupling of an aryl halide dye, for example. For instances where flow cytometry is used for particle characterization, provide said flow data (e.g., LL 195) in the supplemental information.
2. Full characterize the discovered switch molecules and perform controls. The fluorescence enhancement on glucose binding is particularly modest, especially given the promiscuity of boronic acid to bind sugars. Select 3–4 random sequences and perform the same sugar titration to prove that the screen yielded bona fide active molecules. The same should be done for the ATP screening results (address 10-fold difference in fluorescence for these molecules; see Figure S7). Other conventional characterization tactics in the field include reversing the aptamer sequence and/or testing the sequences without the modification. All of these control molecules must be synthesized and evaluated to rule out artifice or non-specific binding at these high ligand and aptamer concentrations.
3. The manuscript devotes undue space to a survey of modeling packages that do not work. Remove all of this text (LL 282-298 + associated SI materials) and focus discussion on topics related directly to the development of the new screening strategy. For example, more attention is due to the use of non-canonical base pairing in switch secondary structure.
4. Provide specific detail on the methods for discovering signaling clusters in the Illumina flow cell. The raw data for cluster selection (Figure 5B) appear to be arbitrary.

Reviewer #3 (Remarks to the Author):

Tom Soh and coworkers report a new massively parallel screening-based strategy that allows the engineering of virtually any aptamer into a molecular switch without requiring any prior knowledge of aptamer structure. The authors exploited a competition-based aptamer switch design where the aptamer strand is labeled with a fluorophore and modified with an anchor portion at 3' end. This portion allows the interaction of the aptamer sequence with a switching strand which is labeled with a quencher and its sequence can be screened and modified to induce a conformational change upon the binding with the target. Using this construct the authors screened thousands of switching strands for the classic ATP-binding aptamer and a newly selected glucose aptamer. The manuscript is very interesting and the approach is novel and potentially applicable to many aptamers for different applications. For the above reasons I support publication of the present manuscript. However, some clarifications/revisions would be needed in order to improve the impact of the paper.

1) The authors claim that: "This makes it possible to evaluate the entire sequence space and identify optimal switch constructs without requiring prior knowledge of the structure or binding sites of the aptamer" (lines 138-139). However, authors selected as a testbed the classic ATP-binding aptamer whose structure is well known from literature. Authors should better explain this choice in a revised manuscript.

2) The authors in the conclusions section claim: "Our approach is generalizable and should therefore help accelerate the generation of functional aptamer-based switches, thereby facilitating the creation of novel biosensors for use in a broad range of application" (lines 454-456). This is fair but authors should also elaborate on possible limitations of this approach. For example, the authors did not investigate too much the effect of the strategy on specificity of the aptamer and on its binding kinetics. This is quite important and should be not too difficult to demonstrate with time-course fluorescence experiments. ATP aptamer is known to have binding kinetics of seconds. If the competing switching strand delays the binding of the target this effect could limit the use of the aptamer, for example, for real-time measurements in vivo. Authors should discuss this issue and, if possible, provide additional data on this.

3) The authors should better describe the features of the proposed strategy from a temporal and economic point of view. What is the total cost to select new switching strands? What is the time required for their selection? At a first sight, the strategy looks expensive (all the oligos are labeled) and requires some experience in handling aptamer binding systems. How this compares to computational approaches? This aspect should be also discussed.

4) The authors should comment on how the unimolecular displacement process between the aptamer sequence and the switching strand affects the nature of their interactions. More specifically, the presence of non-canonical interactions is due to their proximity? If the two strands are not connected by

the anchor domains the pattern of these interactions could be different? This should be better described and discussed.

Reviewer #1: In this study, Yoshikawa et al., reported an Illumina MiSeq-based high-throughput screening strategy to engineer aptamers into molecular sensors. This similar platform has been used by the authors previously to select base-modified aptamers (Ref 24) and to improve the specificity of the aptamers (bioRxiv 10.1101/2021.11.01.466780). Here, the authors further demonstrated that molecular switches can be identified by screening a library of random 10 nucleotide-long switch domain from binding with either an ATP aptamer or a glucose aptamer. The major advantage of this strategy, as compared to conventional rational design strategy, is its throughput and generalizability, as it does not require prior knowledge of aptamer structure, meanwhile, non-natural nucleotide-based aptamer can also be used for the screening. While this is an interesting approach, however, the requirement of a specialized (and expensive) MiSeq instrument can easily prevent the broad usage of this method in general laboratories. More importantly, the relatively small signal-to-noise ratio of these identified sensors (e.g., for glucose, maximally only 1.5-2-fold decrease in fluorescence) further reduce the reviewer's excitement on this approach. Some other comments are listed below.

We thank the reviewer for their thoughtful comments, and have listed these below along with our detailed responses:

1. **“For both ATP and glucose switches, the motif nearby the 5'-end was identified as the most abundant hits. The authors suggested that was surprised (P10), however, that was not surprised to the reviewer because the quencher was modified at this 5'-end of the aptamer, and it has been well known that the fluorescence quenching is highly distance-dependent. It raised the first concern, even though the random switch domain was used, while actually the real “binding site” in the aptamer may be already predetermined by the location of the quencher. Or at least, it will largely prefer that site. That may be also one of the reasons why only a small change in the fluorescence signal was observed, because this 5'-end may not the ideal binding site to engineer a molecular switch. As in the case of ATP aptamer, it may be not close to the binding pocket.”**

We thank the reviewer for this feedback, but we are somewhat confused regarding this comment. The 5'-end motif identified for the ATP aptamer (which we termed m3) was not actually the most abundant, and represented only 4.9% of the top 1,000 sequences. Instead, m1 was the most abundant motif (28.2% of the top 1,000 sequences). Furthermore, we reported in the manuscript that we were not surprised to identify the 5' m3 motif, stating that: *“... we found that the m2 motif was complementary to one of the ATP binding pockets, whereas the m3 motif recognizes the stem of the aptamer—both representing regions that one might target with a rationally-designed SD.”* What we did find surprising was the abundance of the m1 motif, since it occurs in a loop region that is distal to the ATP aptamer stem and the binding site. This region

is also far from the 5' end of the aptamer, and this enrichment is therefore unlikely to be a product of the quencher-proximity bias cited by the reviewer. We would also note that it is difficult to accurately predict the distance between fluorophore and quencher without 3D structural analysis of a given aptamer switch. To avoid confusion, we have edited the manuscript to clarify where in the ATP aptamer sequence each motif binds to.

2. “Following this point, whether this approach can only be used for short-stranded aptamers, e.g., 27 or 30 nt-long strands used in this study? This is important because both natural riboswitches and many synthetic aptamers can be much longer.”

We thank the reviewer for this question. In our method, the only limitation on the length of the aptamer is the ability to synthesize a molecule with a quencher and a ~20–25-nt sequence at its 3' end that can anneal to the switching strand library on the MiSeq flow-cell surface. Companies routinely synthesize modified DNA oligos of up to 200 nt in length using solid-phase synthesis (<https://www.idtdna.com/pages/products/custom-dna-rna/dna-oligos/ultramer-dna-oligos>). Most aptamers and riboswitches fall within this length range, and we therefore believe that the length of the aptamer is unlikely to be a practical limitation for this technology.

3. “Why did the authors choose to use a 10-nucleotide-long switch domain? How will altering the length of this domain affect the switching behavior?”

We opted for a 10-nt switch domain because this would allow us to screen nearly every permutation of the N10 sequence space ($4^{10} = 1,048,576$ possible sequences) in a single experiment using our instrument, thereby conferring more fine-grained insight into the genotype-phenotype relationship. We believe that the optimal length of the switch domain and poly-T linker are likely to be dependent on the specific aptamer being targeted, and these parameters could be interesting to examine in future work. We have added a sentence to the Results and Discussion section clarifying that longer SD regions could be used in this screening method, as well as a short discussion of the exploration of alternative SD and poly-T linker lengths in the concluding paragraphs.

4. “While the authors indicated the SELEX process to identify glucose aptamers, the detailed sequencing result used to identify NNG and the process and results of shortening NNG to obtain NNGmin should be provided. Meanwhile, the target selectivity of this aptamer should be examined.”

We have added details regarding the sequencing results from particle display to **Supplementary Figure 19**, along with secondary structure predictions for the top five candidates and initial binding characterization via flow cytometry. Aptamer truncation is traditionally accomplished by using secondary structure prediction software to identify conserved stem loop regions. After identifying Glu1 as the highest-affinity sequence, we observed that Glu1, Glu3, and Glu4 all shared a similar secondary structure element near the 3' terminus. We thus hypothesized that this 30-nt hairpin was the binding site and compared its binding to the original aptamer. The Results & Discussion section has been updated to describe these points:

*“After three rounds of click-PD screening (**Supplementary Figure 18**), we identified five candidate sequences with the highest copy number in the final-round sequencing data. Of these, we determined the sequence that had the highest affinity for glucose (NNG)... This stem loop structure appeared to be conserved in three of the five candidate sequences, further supporting the importance of this region”*

Supplementary Figure 19: Secondary structure prediction for the five highest copy-number candidate sequences from the PD screen (top) and initial binding validation using flow cytometry (bottom).

Regarding the shortening of the parent aptamer to the minimized NNGmin sequence, we would like to draw the reviewer's attention to **Supplementary Figure 20**, where we show the flow cytometry results comparing the response to glucose before and after minimization. Per the reviewer's suggestion, we have also interrogated the specificity of our glucose ADS constructs with fructose, and note that we do observe switching in response to increasing concentrations of fructose. We were not surprised to observe off-target binding of fructose, given that boronic acids react with any saccharide diol, but we were also pleased to observe lower affinity for this sugar, indicating that glucose is still a primary, preferred binding target for these ADS constructs. We attribute this to the fact that we performed our pre-enrichment with natural DNA, which likely encouraged interactions with the DNA sequence itself and not just the boronic acid moiety. We have added the following to our discussion of the plate-reader validation results:

“We explored the selectivity of our ADS switches by investigating binding to an alternative sugar, fructose (Supplementary Figure 24D). We did observe some off-target binding to fructose, which is not surprising since boronic acid can interact with any saccharide diol and we did not include any counterselection to boost aptamer specificity. However, we were encouraged to observe that our switches exhibited 1.5–2.3-fold lower affinity for fructose compared to glucose, which indicates that glucose is still the preferred primary target for these switch constructs.”

5. “The chemical structures of those boronic acid-modified bases should be provided.”

We apologize for this oversight and have added this chemical structure to **Supplementary Figure 11** as shown below.

Supplementary Figure 11: Chemical structure of the boronic acid-modified dU used in our glucose switch constructs. The modified base is generated through a two-step conjugation of an aminoallyl dU to DBCO-NHS ester followed by phenylboronic acid-azide.

6. “The authors should have a brief explanation of why choosing the current poly T structure and length and also its potential interactions with the aptamers.”

We agree that a discussion of the poly-T linker would strengthen the manuscript. Poly-T linkers are widely used to link aptamers to other moieties such as nucleic acids, proteins, or surfaces, and are particularly advantageous in this context because they can easily be incorporated onto the sequencing flow-cell during the bridge PCR step that occurs during Illumina-based sequencing-by-synthesis. In general, it is reasonable to expect limited interactions of a poly-T linker with the aptamer unless the aptamer itself contains some manner of internal poly-A sequence. The primary design decision we made was regarding the linker length. If the linker is too short, the switching domain may not be able to access the entire aptamer due to spatial constraints. This would clearly limit the ability of the screen to screen switching domains against the entire

aptamer and could lead to non-optimal results. On the other hand, if the linker is too long, the kinetics of the aptamer switch may suffer, potentially creating the need for SDs with length >10 nt. This is because if the linker length is very long, the effective concentration of the switching domain will be lower, such that the binding affinity between the SD and aptamer may have to be increased (see B. Wilson, et al. *Nature Communications* (2019)). Taking these factors into account, and based on the average size of aptamers in the PDB database, we decided to utilize a linker length of 25 nt. While this linker length produced effective results, it would be interesting to investigate the impact of linker and/or SD length on the aptamer switch screen in future work. We have added a short discussion of the poly-T linker length to the manuscript:

“The SD sequence is attached to the anchor strand through a 25-nucleotide poly-T linker, which we believed would have minimal interaction with other DNA strands in the absence of extended stretches of adenosine bases. The length of the linker is crucial; too short of a linker may prevent the SD from accessing the entire aptamer due to spatial constraints, whereas too long of a linker may result in slower kinetics and the requirement of longer SD sequences due to a decreased effective concentration of the SD relative to the aptamer.”

7. “In the introduction, the authors claimed that their switches ‘represent the first example of a base-modified aptamer switch’, which is clearly not true! Many studies have been using modified bases to engineer aptamer sensors, just not name it as “aptamer switch”, see just for example, *Sci. Rep* 2017, 42716 or *Bioconjugate Chem.* 2011, 22, 282-288.

We appreciate the reviewer’s perspective, and have removed this language from the revised manuscript.

8. “P3 ‘engineered riboswitches are also utilized as signaling molecules’, the use of ‘signaling molecules’ can be confusing because of its real biological meaning.”

We agree, and have replaced the words “signaling molecules” with “molecular switches”.

9. “P5, the authors discussed that ‘The “switching strand”... is responsible for endowing the construct with the ability to undergo target binding-induced conformational switching’. The reviewer may not agree with this point. Even without the switching strand, likely these aptamers can still undergo target binding-induced conformational switching and changes.”

Technically, we agree that almost any biomolecule will undergo some sort of conformational change upon target binding. However, our research group has found that leveraging these binding-induced changes to achieve a molecular switch function that modulates an observable signal is a challenging endeavor. The

change in an aptamer's structure upon binding is often very small and insufficient to modulate the distance between, for example, a dye and quencher. Even if an aptamer happens to undergo a large change in conformation upon binding, it could be a challenge to identify the correct locations to place the quencher and fluorophore in that aptamer, since no modeling software can accurately predict the structure of the aptamer upon binding. We have clarified that the switching strand enables the construct to undergo a target binding-induced conformation change *“that results in a change in fluorescent signal.”*

10. “In Page 6 second last sentence, it is also possible that conformational change itself is not enough to induce any change in the fluorescent signal, which should be mentioned.”

We thank the reviewer for the comment. We have added the following text to address this point:

“Although in many cases the ADS constructs will not change signal in response to the target molecule, in successful ADS constructs target binding causes the aptamer strand to undergo a conformational switch that changes the average distance between the fluorophore and quencher, resulting in altered fluorescent signal (Figure 1B).”

11. “Figure S2, why there was a decrease in the fluorescence intensity, because a fluorescently-labeled DNA strand was used? The current figure caption didn't provide any such information.”

We apologize for the confusion. In this experiment, the strand containing a 3' fluorophore is cleaved by the DdeI restriction enzyme, thus resulting in a decrease in fluorescence. We have modified the caption to provide additional information, and have also provided a reference to the Methods section (“Validation of TdT and DdeI enzymes on beads”), where the assay methods are described in detail. The revised SI figure is provided below.

Supplementary Figure 2: Bead-based proof-of-concept experiment for restriction enzyme cleavage by Ddel. A 5'-biotinylated test strand was 3'-labeled with Cy3 and captured onto streptavidin-functionalized magnetic beads. A complementary strand was then annealed on, and the beads were subjected to Ddel enzymatic cleavage. The beads were analyzed via flow-cytometry before and after the enzymatic cleavage. The decrease in fluorescence after the addition of the Ddel indicates successful cleavage of the DNA. See the **Methods** section (“Validation of TdT and Ddel enzymes on beads”) for additional details.

12. “In Figure S3, why the percentage of R3 region signal increased in the first 30 min, and then decreased?”

The gates (R1 and R3) in this figure are arbitrarily designated, and were simply used to provide a frame of reference. The greater value at 30 minutes in the R3 gate is almost certainly due to experimental variance (*e.g.*, small differences in the washing of the beads). The important takeaway from the data should be that the mean fluorescence of the bead population increases substantially at 30 minutes, and does not seem to increase after 60 minutes. We would also like to note that conclusive evidence of the TdT labeling is shown in **Supplementary Figure 4**, which presents an image of the MiSeq flow-cell directly after enzymatic labeling.

13. “As mentioned in my first point, since it was not a surprise that m1 motif can be abundant, the discussion in the P10, ‘and this indicates that the short DNA loop...’ part should be revised.”

As written in our response to the first comment, we believe that the reviewer may have mistaken the m1 motif (which occurs in the small loop of the aptamer) for the m3 motif (which occurs at the 5' end of the aptamer). As such, we believe that this aspect of the manuscript does not need to be revised.

14. “As shown in Figure S7, the fold increase in the fluorescence signal was much reduced as compared to that on the flow cell. It should also be discussed in the main text. Why is that?”

We appreciate the reviewer’s question. These two assays are very different from each other, and it is not necessarily surprising that the values do not precisely match. The plate-reader assay is meant to be quantitative, whereas the MiSeq hardware and software is meant to be more qualitative in nature—this is because its intended use is to simply differentiate between four different fluorophores during sequencing. The differences seen in intensity could be due to experimental factors (*e.g.*, effects due to immobilization or avidity on the flow-cell) or the algorithms used to convert the cluster images into relative fluorescent units (for example, the algorithm may also pick up signal from nearby clusters). The intent of this screening step is to provide relative values that can be used to identify the best switches for further characterization, and not to derive definitive binding measurements. That is why the actual characterization and validation of the switches was done using the plate-reader assay, which is designed for quantitative fluorescent measurements. We have added the following to the manuscript to address this point:

“While the N2A2 system is a powerful tool for screening many aptamer sequences in parallel, the instrument is not necessarily designed for quantitative fluorescent intensity measurements. Rather, the MiSeq hardware is intended to differentiate between four fluorophores (corresponding to each DNA base) during conventional sequencing. Measurements collected from this instrument can potentially be confounded by factors such as avidity effects between neighboring ADS constructs within a cluster, or artifacts introduced by the algorithm that is being used to convert the cluster images into relative fluorescent units (for example, the algorithm may pick up signal from nearby clusters). This means that the screening step is best suited to provide relative binding values between the ADS clusters and to identify the best switches for further characterization.”

15. “Similarly, while atp-7 exhibited the highest affinity, it also exhibited the lowest signal-to-noise ratio. It should be discussed.”

We thank the reviewer for this comment. The relative signal changes observed upon binding will be dependent on the change in distance between the fluorophore and quencher upon target binding. In the case of atp-7, this change in distance is likely less than some of the other aptamer switches, such as atp-4. While this change in distance may be inherent to the aptamer switch, it is also possible that the signal change could

be increased by optimizing the location of the quencher and fluorophore post-screening. We have added a short discussion of this to the manuscript.

16. “‘Monoclonal aptamer particles’ is not a typical phrase, more like a jargon, which should be revised or explained.”

We have clarified this term as follows: “... *monoclonal aptamer particles (i.e. particles containing copies of a single aptamer sequence)*...”. We would like to retain the usage of this term to maintain consistency with our previous publications using the particle display technology.

17. “The format of references 24, 28, 36, 39 should be revised.”

We thank the reviewer for catching these errors and have corrected them in the manuscript.

18. “Some typos: In Figure 1C, ‘High-throughput screening’; Figure S1, ‘Ddel’; In Figure 4C, label of y-axis is missing; ‘Thus, so we carried out the analysis, fully recognizing is limitations (see methods)’. Also, it is a bit confusing what the methods are referring to.”

We thank the reviewer and have corrected these errors. For the statement regarding the limitations of available modeling programs, we intended to highlight the fact that there are no tools readily available to predict the secondary structure of base-modified DNA. We believe that the preceding sentences explain this sufficiently and we have removed the sentence referenced in this comment to avoid any confusion.

Reviewer #2: “Yoshikawa et al. describe a high-throughput approach to identifying novel switch aptamers. Switches are usually discovered in binding selections and then designed for activity through some combination of computational and rational design. These approaches are typically fraught because design rules are incomplete or sometimes wrong. Here, an array of aptamers is generated on an illumina sequencing slide and interrogated for function by hybridizing a quencher strand and monitoring the array for changes in fluorescence after the target molecule is added. The approach is demonstrated for an ATP-sensing switch and a glucose-sensing switch. While the approach is potentially interesting, the data are largely speculative and none of the discovered molecules or reagents used are characterized to a level that supports the conclusions. Publication is premature at this time. I provide major concerns.”

We thank the reviewer for their comments, and have listed these below along with our detailed responses:

1. “None of the reagents or discovered molecules are characterized. Multiple times, methods are presented that assume that click reactions yield only product. While these reactions are efficient, this is dramatic oversimplification and it could have profound implications for the screen. Characterize, for example, the glucose-dye conjugates, the alkyne beads, and the azido phenyl boronic acid products. Routine chemical analysis includes mass spectrometry, liquid chromatography, and/or NMR spectroscopy. Characterizing the boronic acid products will be more difficult. Consider a Suzuki cross-coupling of an aryl halide dye, for example. For instances where flow cytometry is used for particle characterization, provide said flow data (e.g., LL 195) in the supplemental information.”

We thank the reviewer for their helpful suggestions and agree that this characterization is a necessary component for fully validating the non-natural aptamers investigated here. We have used MALDI mass spectrometry to characterize all conjugation steps for the two-step functionalization of the NNGmin sequence, including: 1) the initial sequence containing the amino allyl dU modifications, 2) after reaction with the DBCO-NHS ester, and 3) the final click reaction with the phenylboronic acid. These data are consistent with seven of the eight amino allyl bases being modified with the boronic acid modification. We hypothesize this is likely due to steric hindrance arising from successful conjugation of the DBCO group to the two consecutive amino allyl dU bases near the 5' end of the NNGmin sequence. We believe this ~90% efficiency in attaching the non-natural adducts is suitable to maintain consistency between all the experiments performed. In order to verify conjugation reactions related to the particle display process, we labeled the alkyne magnetic beads with a AF488 azide and confirmed that we see the expected shift in the FITC channel during flow cytometry. This confirms the efficiency of the copper click reaction that was initially used to immobilize the glucose-azide onto the magnetic beads for pre-enrichment. We also

validated the successful conjugation of the glucose-AF647 conjugate used for click-PD by MALDI. The data have been added to the SI, and these experiments are summarized in the text as follows: “We used a variety of experimental methods to characterize the various components and conjugation steps utilized for click-PD, including the two-step boronic acid modification of the DNA, the alkyne beads utilized for pre-enrichment, and the fluorophore-labeled glucose utilized for FACS screening. These results are detailed in **Supplementary Figures 10-17.**”

Supplementary Figure 10: $^1\text{H-NMR}$ characterization of azido phenylboronic acid.

Supplementary Figure 11: Chemical structure of the boronic acid-modified dU used in our glucose switch constructs. The modified base is generated through a two-step conjugation of an aminoallyl dU to DBCO-NHS ester followed by phenylboronic acid-azide.

Supplementary Figure 12: Validation of click chemistry yield using denaturing PAGE. Gel lanes represent 1) Bio-NNGmin 2) Bio-NNGmin after reaction with DBCO-NHS ester, and 3) Bio-NNGmin after reaction with both DBCO NHS ester and phenylboronic acid azide.

Supplementary Figure 13: MALDI-MS characterization of Bio-NNGmin, which contains eight amino allyl dU modifications. Expected mass: 10,026.71 g/mol; observed mass: 10,059.15 g/mol [M-2H+K].

Supplementary Figure 14: MALDI-MS characterization of Bio-NNGmin after reaction with DBCO sulfo-NHS ester (BioNNGmin+DBCO). Expected mass: 12,239.49 g/mol (seven DBCO) and 12,554.86.g/mol (eight DBCO); observed mass: 12,215.12 g/mol [M+Na].

Supplementary Figure 15: MALDI-MS characterization of Bio-NNGmin+DBCO after reacting with azido phenylboronic acid (Bio-NNGmin+BA). Expected mass: 14,323.26 g/mol (eight DBCO + eight BA), 14,102.21 g/mol (eight DBCO + seven BA), 13,786.84 g/mol (seven DBCO + eight BA); observed mass range: 13,691.93–14,079.32 g/mol.

Supplementary Figure 16: MALDI-MS characterization of AlexaFluor 647-labeled glucose conjugate. Expected mass: 1,181.4 g/mol; observed mass: 1,124.06 g/mol.

Supplementary Figure 17: Flow cytometry validation of conjugation to alkyne magnetic beads during pre-enrichment. Beads were monitored in the FITC channel before and after reaction with azide-modified AlexaFluor 488 using CuAAC click chemistry to assess the efficiency of conjugation to bead surfaces.

2. “Full characterize the discovered switch molecules and perform controls. The fluorescence enhancement on glucose binding is particularly modest, especially given the promiscuity of boronic acid to bind sugars. Select 3–4 random sequences and perform the same sugar titration to prove that the screen yielded bona fide active molecules. The same should be done for the ATP screening results (address 10-fold difference in fluorescence for these molecules; see Figure S7). Other conventional characterization tactics in the field include reversing the aptamer sequence and/or testing the sequences without the modification. All of these control molecules must be synthesized and evaluated to rule out artifact or non-specific binding at these high ligand and aptamer concentrations.”

We agree that such control experiments are important, and have included these results in the revised manuscript. We ran five scrambled control sequences with the same base composition as the various ATP aptamer switch domains to ensure that the specific sequence of the switching domain was responsible for conferring structure-switching properties. More specifically, we examined two different scrambled versions of atp-1 and individual scrambled controls for atp-2, atp-4, and atp-6. As expected, no increase in signal was seen with these control strands over the same concentration range, indicating that specific switching domain sequences are required for target-dependent signaling of the ADS construct. These data have been added to the SI (**Supplementary Figure 8**) along with the sequences of the scrambled controls (**Supplementary Table 2**).

Supplementary Figure 8: Raw RFU values from plate-reader binding experiments with scrambled versions of the atp-1, atp-2, atp-4, and atp-6 sequences. Each point represents the mean of quadruplicate experiments, and the error bars represent the standard deviation.

Next, we performed similar control experiments for our glucose ADS constructs. For this comparison, we analyzed derivatives of our various NNGmin-based aptamers (glu1–4) with scrambled SD sequences, natural DNA-based analogs of the NNGmin sequence (NatGmin), or scrambled versions of the NNGmin sequence with the original SD sequence left intact (NNGmin_scr). With the NatGmin constructs, we observed only a small decrease (~10%) in fluorescence for all four switches as glucose concentrations increased. These results suggest that the boronic acid modification is important for efficient glucose recognition and binding-induced displacement of the SD sequence. We also observed a minimal fluorescence response to glucose for both the scrambled SD sequences (~7%) and the NNGmin_scr constructs (~20%). These low levels of non-specific binding are unsurprising due to the formation of boronate esters, which are likely to cause some shift in conformation even when their position in the strand is altered. Despite these low levels of non-specific activity with the scrambled aptamer and SD constructs, the change in fluorescence over the measured concentration range is substantially less than that of the original glu1–4 switches, which yielded a 45–50% signal change. These data have been added to the SI as **Supplementary Figure 24A–C**, along with the sequences of the scrambled controls (**Supplementary Table 4**), and the following description has been added to the discussion:

“In order to assess the specificity of these ADS constructs, we also tested various control versions of the glu1–4 constructs. Specifically, these included constructs incorporating the natural DNA analog of NNGmin, a scrambled version of NNGmin, and scrambled SD sequences. While we observed some modest

glucose response from some of these control sequences, the signal change was less (7–10% for natural DNA and scrambled SD; 20% for scrambled NNGmin) compared to the original glu1-4 switches (45-50% signal change) (**Supplementary Figure 24A-C**). We hypothesize that the higher non-specific activity of the scrambled NNGmin results from the presence of the boronic acid modifications, which encourages folding patterns that are conducive to glucose binding even when their locations are altered relative to the original sequence. This is not surprising, as the strong interaction of glucose and boronic acid will likely perturb interactions between the two strands. However, the approximately 2-fold decrease in signal change magnitude that results from scrambling the modifications indicates that the original locations of the boronic acids are necessary for optimal folding response.”

3. “The manuscript devotes undue space to a survey of modeling packages that do not work. Remove all of this text (LL 282-298 + associated SI materials) and focus discussion on topics related directly to the development of the new screening strategy. For example, more attention is due to the use of non-canonical base pairing in switch secondary structure.”

We appreciate the reviewer’s perspective and have cut this material from the revised manuscript.

4. “Provide specific detail on the methods for discovering signaling clusters in the Illumina flow cell. The raw data for cluster selection (Figure 5B) appear to be arbitrary.”

We thank the author for the comment, and are happy to clarify our process. The pipeline for processing the MiSeq data has been described in our previous publications (references are provided below this response), and the intensity cutoffs that we used are described in the main text of the manuscript. In short, after running our previously published custom code to link the cluster sequences to cluster intensities, clusters with fluorescence < 100 RFU or > 1,000 RFU were removed from the analysis. Clusters with > 30% relative standard deviation between either buffer cycles or replicate target cycles were also excluded. Finally, the clusters were simply sorted based on the ratio of the signal with target to the signal in buffer. We have added references to the Methods section and have also provided a GitHub link within the manuscript that links to the previously-published processing code. The clusters shown in **Figure 5B** are representative images taken of the cluster sequences that were identified by our automated analysis pipeline, and are simply provided as representative images.

References for MiSeq data processing

- (24) Wu, D.; Feagin, T.; Mage, P.; Rangel, A.; Wan, L.; Li, A.; Coller, J.; Eisenstein, M.; Pitteri, S.; Soh, H. T. Automated Platform for High-Throughput Screening of Base-Modified Aptamers for Affinity and Specificity. *bioRxiv* **2020**, 1–23. <https://doi.org/10.1101/2020.04.25.060004>.
- (25) Yoshikawa, A. M.; Wan, L.; Zheng, L.; Eisenstein, M.; Soh, H. T. A System for Multiplexed Selection of Aptamers with Exquisite Specificity without Counter-Selection. *PNAS* **2022**, 2021.11.01.466780. <https://doi.org/10.1101/2021.11.01.466780>.

Reviewer #3: “Tom Soh and coworkers report a new massively parallel screening-based strategy that allows the engineering of virtually any aptamer into a molecular switch without requiring any prior knowledge of aptamer structure. The authors exploited a competition-based aptamer switch design where the aptamer strand is labeled with a fluorophore and modified with an anchor portion at 3’ end. This portion allows the interaction of the aptamer sequence with a switching strand which is labeled with a quencher and its sequence can be screened and modified to induce a conformational change upon the binding with the target. Using this construct the authors screened thousands of switching strands for the classic ATP-binding aptamer and a newly selected glucose aptamer. The manuscript is very interesting and the approach is novel and potentially applicable to many aptamers for different applications. For the above reasons I support publication of the present manuscript. However, some clarifications/revisions would be needed in order to improve the impact of the paper.”

We thank the reviewer for their thoughtful comments and have addressed them below.

1) “The authors claim that: ‘This makes it possible to evaluate the entire sequence space and identify optimal switch constructs without requiring prior knowledge of the structure or binding sites of the aptamer’ (lines 138-139). However, authors selected as a testbed the classic ATP-binding aptamer whose structure is well known from literature. Authors should better explain this choice in a revised manuscript.”

We thank the reviewer for this thoughtful comment. We actually selected the ATP aptamer as an initial proof-of-concept demonstration because it is one of the few aptamers with an NMR-resolved structure, which enabled us to better evaluate the nature of the interactions between the screened switching domains and the parent aptamer itself. For example, we were able to determine with reasonable confidence whether the switching domain motifs that were identified in the screen were interacting with a hairpin, binding pocket, or loop region. We have revised the manuscript to explain our initial choice of this well-studied ATP aptamer in the Results and Discussion. That being said, our subsequent results with a newly-discovered (and structurally uncharacterized) glucose aptamer offer confirmation that no such *a priori* knowledge is required to successfully employ this approach.

2) “The authors in the conclusions section claim: ‘Our approach is generalizable and should therefore help accelerate the generation of functional aptamer-based switches, thereby facilitating the creation of novel biosensors for use in a broad range of application’ (lines 454-456). This is fair but authors should also elaborate on possible limitations of this approach. For example, the authors did not investigate too much the effect of the strategy on specificity of the aptamer and on its binding kinetic. This is quite important and should be not too difficult to demonstrate with time-course fluorescence experiments. ATP aptamer is known to have binding kinetics of seconds. If the competing switching strand delays the binding of the target this effect could limit the use of the aptamer, for example, for real-time measurements in vivo. Authors should discuss this issue and, if possible, provide additional data on this.”

Since the switching strand is essentially creating a competing binding reaction to the target, it is expected that the kinetics and binding affinity will both decrease relative to the parent aptamer. As suggested by the reviewer, we have performed an analysis of the binding kinetics for ATP aptamer switches atp-4, atp-5, and atp-6. These sequences were chosen because they were the highest-affinity representatives of the three major SD motifs that we identified in this work (m1, m2, and m3). Although their kinetics were slightly slower than reported for the parent ATP aptamer, equilibrium was still achieved relatively quickly (<10

seconds). We have added a short discussion of the kinetics, and have included the data as **Supplementary Figure 9** (data is shown below). There are several other potential limitations of our switches. For example, the location of the quencher and fluorophore may not be optimal to maximize the signal change, although this could be corrected post-screen to optimize the signal change of the switches. Additionally, since the aptamer and switching strand are not covalently attached, the construct may not be as stable as a covalently bonded intramolecular displacement strand aptamer switch. However, the switch could be synthesized as a single strand as long as it is within the length limitations of solid-phase oligonucleotide synthesis.

Supplementary Figure 9: Kinetic binding analysis of atp-4, atp-5, and atp-6 switches. Kinetic binding behavior was examined at various concentrations of ATP. Signals are normalized to the final RFU values. Each data point represents the mean of three independent experiments and the error bars represent a single standard deviation.

3) “The authors should better describe the features of the proposed strategy from a temporal and economic point of view. What is the total cost to select new switching strands? What is the time required for their selection? At a first sight, the strategy looks expensive (all the oligos are labeled) and requires some experience in handling aptamer binding systems. How this compare to computational approaches? This aspect should be also discussed.”

We believe that our system is comparably priced while also being considerably faster (and more efficient) than conventional methods to create aptamer switches (excluding the initial capital cost of acquiring the MiSeq sequencer). Since the same switching strand library can be used for any parent aptamer, most of the reagents (such as the oligo library, enzymes, etc) can be purchased once and used to screen many different aptamers. The primary cost of the screen is associated with purchasing the HPLC-purified, quencher-labeled parent aptamer (~\$250), the MiSeq reagent kit (~\$900), and the Cy3-labeled ddUTP (~\$75), which totals approximately \$1,225. This initial investment makes it possible to evaluate millions of aptamer switch constructs. In contrast, the same cost would only be sufficient to order and test ~5 aptamer switches that were designed using conventional heuristics. We believe that this scale is inadequate for the task at hand; in our experience, it can take around a dozen candidates and multiple design iterations to successfully create effective aptamer switches using these methods. Since multiple design iterations are likely to be required, screening these constructs can take many months, while our screen only takes ~2 days to run. Our approach offers the clear advantage of making it possible to rapidly identify aptamer switches from millions of different designs, rather than slowly iterating on a small number of rationally designed constructs, which both increases the odds of success while also enabling the unexpected discovery of effective switch sequences that would not be predictable using conventional design heuristics. We have added a brief discussion on these aspects of the screen to the conclusion of the manuscript.

4) “The authors should comment on how the unimolecular displacement process between the aptamer sequence and the switching strand affects the nature of their interactions. More specifically, the presence of non-canonical interactions is due to their proximity? If the two strands are not connected by the anchor domains the pattern of these interactions could be different? This should be better described and discussed.”

We thank the reviewer for this comment. As discussed in comment #2 above, we have added a short description of the kinetics and thermodynamics of the switch construct compared to the parent aptamer. In our previous work with intramolecular displacement strand aptamer switches, we found that the displacement strands functioned when covalently linked to the aptamer as well as in solution (B. Wilson,

et al. *Nature Communications* (2019)). Since the constructs are very similar in this work (the two strands are hybridized together rather than covalently linked), we expect these findings to hold true. However, it is always possible that linking the displacement strand can have unexpected consequences and it is not unreasonable to expect that in some cases unique structural motifs could be stably formed. We believe that this would need to be investigated on a case-by-case basis and would be dependent on the specific parent aptamer as well as the aptamer switch. Due to challenges of actually determining the binding mechanism of aptamer switches, which would likely require X-ray crystallography experiments, we believe that a detailed investigation remains outside the scope of this work.

REVIEWER COMMENTS

Reviewer #1 (Remarks to the Author):

The authors have done a good job in revision. My original concerns have been well addressed. Even though the identified sensors still exhibit moderate performance, the development of this high-throughput system can be still inspiring and important for the future engineering of optimal aptamer switch sensors. I will be happy to support the acceptance of this current study.

Reviewer #2 (Remarks to the Author):

Yoshikawa et al. describe a high-throughput approach to identifying novel switch aptamers. Switches are usually discovered in binding selections and then designed for activity through some combination of computational and rational design. These approaches are typically fraught because design rules are incomplete or sometimes wrong. Here, an array of aptamers is generated on an illumina sequencing slide and interrogated for function by hybridizing a quencher strand and monitoring the array for changes in fluorescence after the target molecule is added. The approach is demonstrated for an ATP-sensing switch and a glucose-sensing switch. While the approach is potentially interesting and revisions have addressed some of the prior concerns, the added data do not inspire confidence that the platform is ready for publication. Publication remains premature at this time.

MAJOR CONCERNS

1. Reagent characterization is low quality, incomplete, or incorrect for key synthons in this study. For example, the MALDI MS characterization of the modified oligos (S14, S15) is poor. The peaks are broad with one peak stretching almost 4 kDa (!). The accompanying data analysis is also incorrect. One cannot assign ion complexes with this limited resolution (see speculative M+K-2H assignment of the modified Bio-NNGmin). The dye-labeled glucose (structure not shown) was analyzed by MALDI-MS as well (S16) and the expected mass is nowhere to be found in the spectrum. None of these data are convincing that the key molecules used in the high-throughput screen or subsequent validation experiments were as claimed. Indeed, they only intensify skepticism.

2. The discovered glucose-responsive sequences are extremely weak and appear to be beyond the detectable limits of the binding assays used for characterization. For example, the binding curves (S24) are so shallow that fluorescence difference is within the reported uncertainty. Binding to fructose (a potential interference) is more robust than to glucose, indicating that binding may be sequence independent. In all, these data do not substantiate the claim that glucose-responsive sequences are being identified by the platform.

MINOR CONCERNS

1. The language throughout the manuscript is overly optimistic and almost feels like an advertisement for a product. Rewrite the abstract to define the key quantitative discoveries of the work. Remove speculative or overly dramatic descriptions of the underlying technology and focus on an objective analysis of the findings.

Reviewer #3 (Remarks to the Author):

For what concerns my questions/clarifications the authors have answered them in a comprehensive way and I thus confirm my support to the publication of the paper in the present form.

Reviewers #1 and #3 were satisfied with the current state of the revised manuscript, but Reviewer #2 expressed a number of concerns that we have addressed in a point-by-point fashion below.

Reviewer #2:

MAJOR CONCERNS

- 1. Reagent characterization is low quality, incomplete, or incorrect for key synthons in this study. For example, the MALDI MS characterization of the modified oligos (S14, S15) is poor. The peaks are broad with one peak stretching almost 4 kDa (!). The accompanying data analysis is also incorrect. One cannot assign ion complexes with this limited resolution (see speculative M+K-2H assignment of the modified Bio-NNGmin). The dye-labeled glucose (structure not shown) was analyzed by MALDI-MS as well (S16) and the expected mass is nowhere to be found in the spectrum. None of these data are convincing that the key molecules used in the high-throughput screen or subsequent validation experiments were as claimed. Indeed, they only intensify skepticism.**

We thank the reviewer for raising these important points. To improve the quality of MS spectra, we have re-analyzed the oligonucleotides by liquid chromatography electrospray ionization tandem mass spectrometry (LC-ESI-MS), which greatly improved the resolution of the spectra. ESI-MS analysis has improved accuracy when measuring higher molecular weight oligos compared to the matrix-assisted laser desorption/ionization mass spectrometry (MALDI-MS) analysis utilized previously. Additionally, we have performed high-resolution mass spectroscopy (HRMS) and HPLC analysis of the AlexaFluor 647 alkyne dye (AF647), the glucose-AF647 conjugate, and sulfo-cy5 alkyne.

To begin, we characterized the dye-labeled glucose in detail (**Supplementary Figures 13, 14, and 15**). The original calculated mass was inferred from the molecular weight for AlexaFluor 647 NHS ester provided by the vendor, because the vendor would not provide the molecular weight or structure of the AlexaFluor 647 alkyne. To obtain the correct mass of Alexa Fluor 647 alkyne, we analyzed the compound by HRMS and HPLC, and the mass of Alexa Fluor 647 alkyne is identical to that of sulfo Cy5 alkyne (**Supplementary Figures 13 and 14**), $[C_{35}H_{41}N_3O_7S_2-H^+]$ calculated $m/z = 678.2313$, observed $m/z = 678.2315$). Additionally, the retention time of Alexa Fluor 647 alkyne is very similar to that of sulfo Cy5 alkyne (Figure 4 and 5 insets). Chromatography conditions are provided in **Supplementary Tables 5 and 6**. Collectively these results suggested that Alexa Fluor 647 alkyne and sulfo Cy5 alkyne are likely to be the same compound apart from possibly bearing different counter ions. We would like to note that the actual structure of the fluorophore is not critical to the characterization of the reagents developed in this study, since the dye was only utilized during the Particle Display selection of the glucose aptamers. Unlabeled glucose was used during the switch screen and subsequent aptamer switch characterization via plate-reader. Finally, we would like to note that the AlexaFluor 647 alkyne dye product has been used extensively within existing peer reviewed literature.

Supplementary Figure 13: HRMS spectrum of AlexaFluor 647 alkyne, inset: chromatogram of AlexaFluor 647 alkyne. $[C_{35}H_{41}N_3O_7S_2-H^+]$ calculated $m/z = 678.2313$, observed $m/z = 678.2315$.

Supplementary Figure 14: HRMS spectrum of sulfo Cy5 alkyne, inset: chromatogram of sulfo Cy5 alkyne. $[C_{35}H_{41}N_3O_7S_2-H^+]$ calculated $m/z = 678.2313$, observed $m/z = 678.2315$.

We further analyzed dye-labeled glucose by HRMS, and proposed the most probable structure of the compound (Figure 6, $[C_{49}H_{68}N_6O_{16}S_2-H^+]$ calculated $m/z = 1059.4060$, observed $m/z = 1059.4065$). These new results are included in the revised supporting information.

Supplementary Figure 15: HRMS spectrum of AlexaFluor 647-labeled glucose. $[C_{49}H_{68}N_6O_{16}S_2-H^+]$ calculated $m/z = 1059.4060$, observed $m/z = 1059.4065$.

Both the starting DNA aptamer material (Bio-NNGmin) and the DBCO-conjugated intermediate material (BioNNGmin+DBCO) also showed the expected masses in our ESI-MS spectra (**Supplementary Fig. 20** and **21**). However, our ESI-MS analysis of the final modified sequence BioNNGmin+BA identified seven boronic acid modifications, rather than the expected eight (**Supplementary Fig. 22**). Our hypothesis is that this resulted from steric hindrance induced by the secondary structure of BioNNGmin+DBCO, which prevented one of the DBCO groups from reacting with boronic acid. We are unsure which position remained unconjugated, but believe that identifying this position is outside of the scope of this work, and would like to emphasize that this does not impact the broader validity of our screening platform. Nevertheless, these results do highlight the importance of performing careful confirmatory analysis of the modification state of the molecules identified using our platform. Our new ESI-MS data from this analysis are shown below, and in the SI of the revised manuscript.

Supplementary Figure 20. **a)** ESI-MS spectrum of Bio-NNGmin and **b)** deconvoluted ESI-MS spectrum of Bio-NNGmin. Calculated mass: 10,026.71; observed mass: 10,031.5.

Supplementary Figure 21. **a)** ESI-MS spectrum of Bio-NNGmin after reaction with DBCO sulfo-NHS ester (BioNNGmin+DBCO), and **b)** deconvoluted ESI-MS spectrum of BioNNGmin+DBCO. Calculated mass: 12,561.46; observed mass: 12,555.5.

Supplementary Figure 22. a) ESI-MS spectrum of Bio-NNGmin+BA, and b) deconvoluted ESI-MS spectrum of Bio-NNGmin+BA. Calculated mass: 14,442.6 (eight DBCO groups conjugated), 14,207.5 (seven DBCO groups conjugated); observed mass:14,264.5 (M-BA-3H+Na+K, calculated mass 14,266.5).

2. **The discovered glucose-responsive sequences are extremely weak and appear to be beyond the detectable limits of the binding assays used for characterization. For example, the binding curves (S24) are so shallow that fluorescence difference is within the reported uncertainty. Binding to fructose (a potential interference) is more robust than to glucose, indicating that binding may be sequence independent. In all, these data do not substantiate the claim that glucose-responsive sequences are being identified by the platform**

We thank the reviewer for the comment, but we believe that the reviewer may have misinterpreted **Supplementary Figure 24** (which is **Supplementary Figure 26** in the revised

manuscript) as evidence that the glucose-responsive sequences are extremely weak. However, panels A–C in **Supplementary Figure 24** show negative control sequences, which would be expected to not show significant binding to glucose. **Supplementary Figure 24A** shows a natural analog of the modified glucose aptamer that should not bind glucose since it does not contain the boronic acid modifications that were selected for during the Click-PD aptamer selection. **Supplementary Figure 24B** and **C** show two different scrambled sequence controls (one of the switching domain, and the other of the aptamer), which act as controls to demonstrate that the specific sequence of the aptamer switch is necessary for it to function. These controls were conducted to account for experimental artifacts resulting from effects such as non-specific binding between DNA and the target, and actually confirm that our non-natural glucose aptamers bind glucose specifically in a sequence-dependent manner. That being said, we appreciate that the nomenclature used in **Supplementary Figure 24** is somewhat confusing and may have been the source of the misunderstanding, and we have added more information to clarify that they are negative controls that exhibit weak or no binding to glucose.

Upon careful examination of the aptamer binding towards fructose and glucose, we believe that the two targets exhibit similar binding affinity and that their dissociation constants do not differ significantly. As we indicated in the manuscript, since no negative selection was conducted against fructose, it is not surprising to see some cross-reactivity due to the structural similarity between glucose and fructose. We have revised the manuscript with the below text to make clear that the aptamers also bind fructose.

“We found that the ADS switches bound fructose and glucose with similar affinities. The significant off-target binding to fructose that we observed is not surprising, due to its close structural similarity to glucose and the fact that boronic acids can interact with any saccharide diol. It is likely that the use of a counterselection step against fructose during the selection process could have increased aptamer specificity towards glucose.”

MINOR CONCERNS

- 1. The language throughout the manuscript is overly optimistic and almost feels like an advertisement for a product. Rewrite the abstract to define the key quantitative discoveries of the work. Remove speculative or overly dramatic descriptions of the underlying technology and focus on an objective analysis of the findings.**

We thank the reviewer for the comment and have attempted to revise the language in the manuscript to address the concerns.